# FAIR-Calib: Frontier-Aware Instability-Reweighted Calibration for Post-Training Quantization of Diffusion Large Language Models

Haoyu Huang[1]   Linlin Yang[2][‡]   Sheng Xu[3]   Boyu Liu[4]   Guodong Guo[5]   Zhongqian Fu[6][†]   Hang Zhou[6]
Baochang Zhang[4][7][†]

## Abstract

Diffusion Large Language Models (dLLMs) refine tokens iteratively but commit them irreversibly, leading to a "stability lag" where early decisions remain fragile even after being written. We reveal that Post-Training Quantization (PTQ) error easily flips these borderline decisions at the write frontier, which are then permanently locked in and amplified. To address this, we propose *Frontier-Aware Instability-Reweighted Calibration* (*FAIR-Calib*), a two-stage PTQ framework for dLLMs. Stage I probes a full-precision teacher to estimate a position prior that combines frontier hits and masked-stage reliability. Stage II performs off-policy, layer-wise calibration by minimizing a reweighted hidden-state MSE, effectively prioritizing the protection of fragile frontier states without requiring expensive end-to-end diffusion rollouts. We further theoretically justify our weighted objective as a surrogate for output KL divergence. Empirically, FAIR-Calib consistently outperforms state-of-the-art baselines on LLaDA and Dream (W4A4), significantly reducing frontier decision flips and suppressing post-commit mismatches across diverse benchmarks.

## 1. Introduction

Transformer-based large language models have achieved remarkable generalization and instruction-following abilities

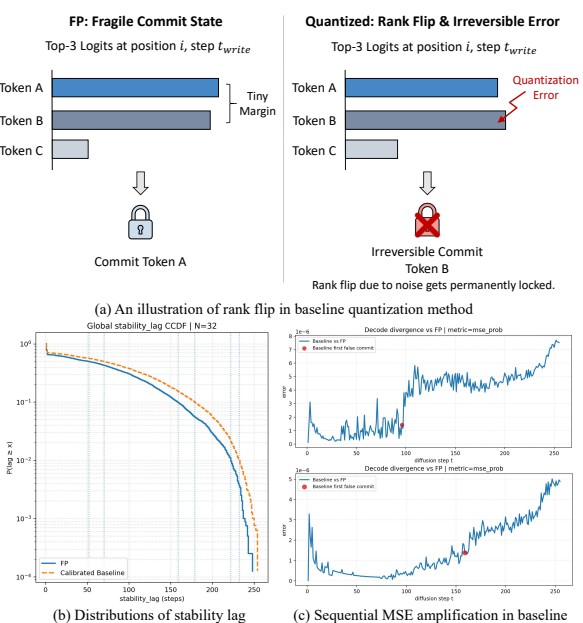

(a) An illustration of rank flip in baseline quantization method

(b) Distributions of stability lag

(c) Sequential MSE amplification in baseline

*Figure 1.* **(a)** Schematic: naive quantization perturbs relative logits and permanently commits the wrong token, highlighting a key failure mode of current quantization methods under diffusion decoding. **(b)** Complementary cumulative distribution function (CCDF) of the stability lag $\delta_{lag}$ in the generation region ($N_{samples} = 32$). Although most positions stabilize shortly after commit, the heavy tail indicates *fragile commit states* that keep oscillating post-commit, showing that **commitment $\neq$ stabilization**. The calibrated baseline exhibits an even heavier tail than FP, implying that standard calibration does not remove fragility. **(c)** Decode divergence w.r.t the FP (metric: $\mathrm{mse\_prob}$) as a function of diffusion step. The baseline shows progressive, step-wise amplification of error once a false commit occurs (red marker), indicating that small local perturbations can trigger sustained divergence across subsequent steps.

[†]Project lead. [‡]Corresponding author. [1]National College for Excellent Engineers, Beihang University, Beijing, China [2]State Key Laboratory of Media Convergence and Communication, Communication University of China, Beijing, China [3]Independent researcher [4]School of Artificial Intelligence, Beihang University, Beijing, China [5]Ningbo Institute of Digital Twin, Eastern Institute of Technology, Ningbo, China [6]Huawei Noah's Ark Lab, China [7]Artificial Intelligence Research Center, Lobachevsky State University, Nizhny Novgorod 603022, Russia. Correspondence to: Linlin Yang <lyang@cuc.edu.cn>.

*Proceedings of the $43^{rd}$ International Conference on Machine Learning*, Seoul, South Korea. PMLR 306, 2026. Copyright 2026 by the author(s).

at the scale of tens to hundreds of billions of parameters, as exemplified by the LLaMA (Touvron et al., 2023) and Qwen (Yang et al., 2025) model families.

Recently, diffusion large language models (dLLMs) have emerged as a promising alternative to autoregressive decoding, offering iterative refinement and flexible infilling by initializing an entire response sequence upfront and denoising it with bidirectional attention over multiple steps (Nie et al., 2025; Zhu et al., 2025; Ye et al., 2025). This iter-

ative masked refinement is conceptually related to earlier non-left-to-right decoding paradigms (Ghazvininejad et al., 2019; Stern et al., 2019; Chang et al., 2022). However, such multi-step global refinement substantially increases inference-time compute and memory footprints, making post-training quantization (PTQ) crucial for practical deployment (Frantar et al., 2022; Frantar & Alistarh, 2023; Xiao et al., 2023; Lin et al., 2024; Ashkboos et al., 2024; Sun et al., 2024).

However, quantizing dLLMs is not a straightforward extension of autoregressive PTQ: Lin et al. (2025) systematically transferred classic low-bit PTQ from autoregressive LLMs to dLLMs and found that naive transfer degrades notably on challenging reasoning tasks. We attribute this brittleness to a diffusion-specific inference mechanism: dLLM decoding proceeds by repeatedly predicting token distributions for all positions and *unmasking* a subset of mask positions into concrete tokens, reducing the mask set step by step. We refer to this irreversible write as a *commit*. This *irreversibility* means that once a token is written, it becomes part of the conditioning context and cannot be revised, even if the model's posterior belief about that position continues to evolve. Consequently, the decoding process becomes particularly brittle under perturbations: as illustrated in Figure 1(**a**), quantization perturbations can easily flip a borderline decision at the write frontier, creating an error that is **permanently locked in**.

We trace this brittleness to a fundamental mismatch: *commitment $\neq$ stabilization*. As visualized in Figure 1(**b**), even in full precision, many positions exhibit a significant *stability lag $\delta_{\text{lag}}$*. We define $\delta_{\text{lag}}$ as the number of diffusion steps after the first irreversible commit until the model's top-1 prediction becomes consistent with the final decoded token for all subsequent steps. This means that many positions continue to oscillate in their top-1 prediction long after being committed. The heavy tail of this distribution reveals a non-negligible subset of *fragile commit states*, where decisions remain context-sensitive and can keep oscillating post-commit. Standard calibration in PTQ methods exacerbates this issue, prolonging the instability and exposing more positions to the irreversible flips described above. Crucially, these locked-in flips do not remain isolated; instead, they can lead to severe degradation in generation quality. Because the incorrect token is fixed as context, it forces the model to refine subsequent tokens based on the error. Figure 1(**c**) confirms this trajectory: once a false commit occurs at a fragile frontier (red marker), the divergence from the teacher does not vanish but undergoes a **progressive, step-wise amplification** across subsequent refinement steps, severely degrading generation quality.

To address these challenges, we propose the ***Frontier-Aware Instability-Reweighted Calibration*** (***FAIR-Calib***) frame-

work for dLLM quantization. Our framework consists of two synergistic stages: (i) *Teacher Probing*: We utilize the full-precision teacher to estimate a position-aware prior. This prior uniquely integrates frontier irreversibility (up-weighting positions at commit time) and masked-stage reliability (accounting for teacher confidence). We show that this prior is largely mechanism-driven and exhibits robust cross-corpus transferability. (ii) Off-policy Weighted Calibration: We perform efficient layer-wise hidden-state alignment using the estimated weights. By employing a teacher-forcing surrogate, FAIR-Calib avoids expensive end-to-end diffusion rollouts while effectively stabilizing the write frontier. Empirically, FAIR-Calib significantly reduces write-step decision flips and post-commit mismatches including both "mean-disagree" and "never-agree" cases. Furthermore, our method successfully mitigates the sequential error amplification typically triggered by false commits, as evidenced by improved probability-MSE traces. Our major contributions in this paper are summarized as:

- We identify and quantify brittleness in dLLM decoding induced by *irreversible commit* under *fragile commit states*, where low-bit quantization flips borderline write decisions and the resulting errors are locked in and amplified across refinement steps.

- We propose **FAIR-Calib**, a two-stage PTQ framework for dLLMs (Figure 2): Stage I probes an FP teacher to estimate a *frontier-aware, reliability-gated* position prior, and Stage II performs *off-policy* layer-wise teacher-forcing calibration via a weighted hidden-state MSE, avoiding expensive diffusion rollouts.

- We justify an additive time$\times$position weighting and its weighted hidden-state MSE surrogate under mild assumptions, and empirically show consistent W4A4 gains on Dream/LLaDA across diverse benchmarks, with fewer teacher-forced commit-step flips, reduced post-commit mismatch, and suppressed error amplification.

## 2. Related Works

### 2.1. Diffusion Language Models

Diffusion models were generalized to discrete state spaces via denoising diffusion over categorical variables (Austin et al., 2021). Subsequent works explored diffusion-style text generation by iterative denoising of token sequences or latent representations, enabling non-left-to-right generation with global revision (Li et al., 2022; Gong et al., 2022). This refinement view is also related to earlier iterative decoding paradigms that repeatedly revise low-confidence positions (Ghazvininejad et al., 2019; Stern et al., 2019; Chang et al., 2022). More recently, diffusion *large* language models

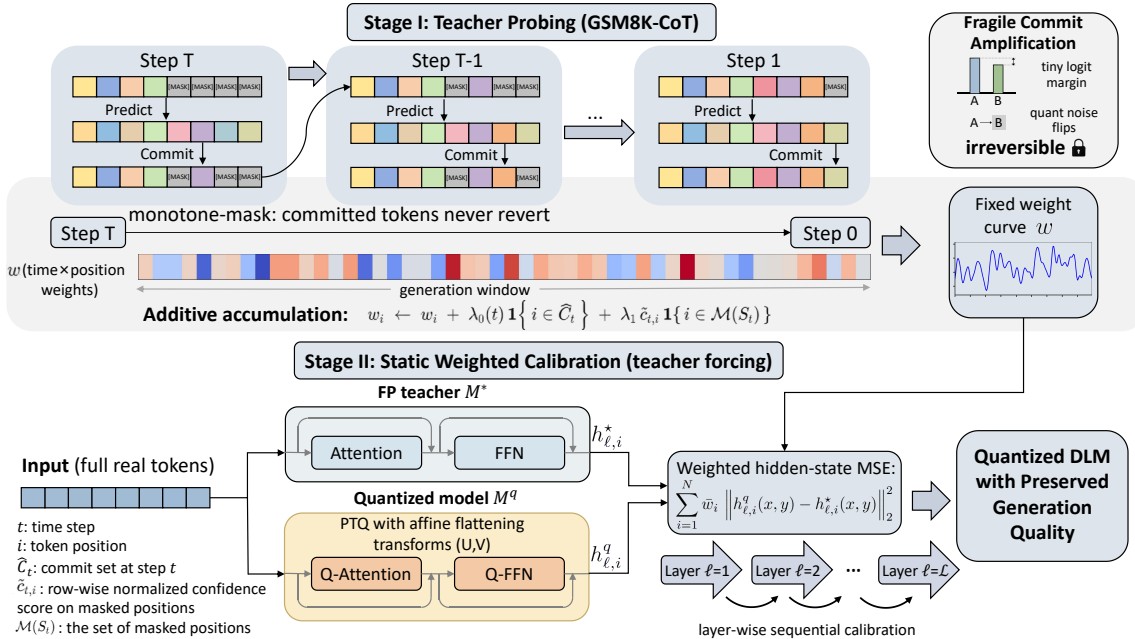

*Figure 2.* **FAIR-Calib overview. Stage I** probes the FP teacher to estimate a fixed position prior $\bar{w}$ that highlights irreversible commit positions and masked-stage reliability. **Stage II** performs layer-wise teacher-forcing calibration with a $\bar{w}$-weighted hidden-state MSE to obtain a W4A4 model without diffusion rollouts.

scale masked refinement to Transformer LLMs by initializing an answer window with masks and denoising it with bidirectional attention over multiple steps (Nie et al., 2025; Zhu et al., 2025; Ye et al., 2025). While enabling flexible infilling, their stepwise *commit* and long-horizon refinement increase inference cost and introduce new brittleness modes for compression such as PTQ.

## 2.2. Post-training Quantization for Large Language Models

Post-training quantization (PTQ) compresses pretrained LLMs by quantizing weights and/or activations with a small calibration set (Zhu et al., 2024). Reconstruction-based PTQ explicitly minimizes layer-wise output/hidden discrepancies, e.g., GPTQ-style second-order updates (Frantar et al., 2022). For low-bit joint weight–activation quantization (e.g., W4A4), distribution mismatch and outliers are key bottlenecks, motivating activation smoothing (Xiao et al., 2023), rotation-based conditioning (Ashkboos et al., 2024), and affine flattening transforms (Sun et al., 2024), along with complementary advances such as activation-aware scaling/clipping and system-oriented stacks (Lin et al., 2024; Shao et al., 2023; Yao et al., 2022; Tseng et al., 2024). Most PTQ methods are developed under autoregression and do not account for diffusion decoding with *irreversible commits*; our work targets this gap.

# 3. Method

## 3.1. Preliminaries

### 3.1.1. PTQ WITH AFFINE FLATTENING TRANSFORMS

We follow a standard post-training quantization (PTQ) setup, quantizing weights and activations to low bit-width without updating pretrained FP weights. In addition, we adopt the same *layer-wise learnable affine transformation* design as FlatQuant (Sun et al., 2024) to flatten weight/activation distributions before applying uniform quantization.

Let $\mathcal{Q} = \{q_{\min}, \dots, q_{\max}\}$ be the integer grid determined by bit-width $b$. Given a scale $s > 0$ and an (optional) zero-point $z$, the quantizer maps a real-valued variable $u$ to

$$\bar{u} = \text{clip}\left(\left\lfloor \frac{u}{s} + z \right\rceil, q_{\min}, q_{\max}\right), \qquad Q(u) = s(\bar{u} - z), \tag{1}$$

where $\lfloor \cdot \rceil$ denotes rounding-to-nearest and $\text{clip}$ clamps to the valid integer range. In this paper, we use symmetric quantization and set $z = 0$.

For each linear layer $y = Wx$, we introduce an invertible affine reparameterization

$$y = U^{-1} \widetilde{W} \tilde{x}, \qquad \widetilde{W} = UWV^{-1}, \ \tilde{x} = Vx, \tag{2}$$

and compute the quantized output as

$$y^q = U^{-1}\left(Q_W(\widetilde{W}) \, Q_X(\tilde{x})\right), \tag{3}$$

where $Q_W$ and $Q_X$ are uniform affine quantizers (with their own scales and, if used, zero-points).

We calibrate the quantization-related parameters for each layer on a small calibration set $\mathcal{D}$, while keeping pretrained weights frozen. Denoting the FP and quantized layer mappings by $F_\ell^\star(\cdot)$ and $F_\ell^q(\cdot; \theta_\ell)$, respectively, we solve a sequential layer-wise reconstruction problem:

$$\min_{\theta_\ell} \mathbb{E}_{x \sim \mathcal{D}} \left[ \| F_\ell^q(x; \theta_\ell) - F_\ell^\star(x) \|_2^2 \right]. \quad (4)$$

In our method, this generic reconstruction loss is instantiated as a *position-weighted hidden-state MSE* that emphasizes fragile commit positions (Section 4).

### 3.1.2. MASKED DIFFUSION DECODING AS A MARKOV CHAIN

Let $\mathcal{V}$ be the vocabulary and $N$ the full sequence length. We denote the prompt length by $N_p$ and the generation window (answer region) by indices $\mathcal{G} = \{N_p + 1, \ldots, N\}$. We index reverse diffusion from $t = T$ (fully masked) to $t = 0$ (final). A masked diffusion decoder maintains a partially observed state

$$S_t \equiv X^{(t)} \in (\mathcal{V} \cup \{[\texttt{MASK}]\})^N, \quad (5)$$

where some positions are concrete tokens and others are [MASK]. Let $\mathcal{M}(S_t) = \{i : S_t[i] = [\texttt{MASK}]\}$ be the set of masked positions.

Given state $S_t$, a model $M$ (FP teacher $M^\star$ or quantized model $M^q$) produces per-position logits $z_{M,i}(S_t) \in \mathbb{R}^{|\mathcal{V}|}$ and categorical distributions

$$p_{M,i}(\cdot \mid S_t) = \text{softmax}(z_{M,i}(S_t)). \quad (6)$$

A decoding policy selects a commit set $C_t \subseteq \mathcal{M}(S_t)$ and writes tokens at committed positions. We emphasize that dLLM decoding is not necessarily "posterior sampling"; it is an algorithmic transition rule.

**Initialization and $p(S_T)$.** The initial state $S_T$ is typically prompt + fully masked generation window:

$$S_T = [x_{1:N_p}, \underbrace{[\texttt{MASK}], \ldots, [\texttt{MASK}]}_{N - N_p}], \quad (7)$$

which induces a degenerate initial distribution

$$p(S_T) = \delta(S_T = \bar{S}_T), \quad (8)$$

i.e., a Dirac delta at the deterministic initialization $\bar{S}_T$.

Quantization-induced local noise propagates and scales within the Markov decoding chain, demanding spatio-temporally aware calibration.

### 3.2. Overview of FAIR-Calib

We propose a two-stage post-training quantization framework tailored for diffusion language models with irreversible commits. It separates *where errors are amplified* from *how calibration is performed*.

**Stage I: teacher probing.** We run a small number of full-precision teacher rollouts *under stochastic (random) commits* and construct a fixed position prior $\bar{w}$ over the generation window. This choice of random commits is deliberate: it aligns the probing process with the randomized masking mechanisms used during dLLM training (Nie et al., 2025) and provides a policy-agnostic coverage of the masked state space, ensuring that $\bar{w}$ reflects the model's intrinsic structural sensitivity. The prior combines (i) a *frontier-hit* signal that marks when a position is committed (irreversible), and (ii) a *masked-stage teacher-reliability* signal computed from the teacher distribution while the position remains masked. Weights are additively accumulated across diffusion steps and then window-aligned/normalized.

**Stage II: static weighted calibration.** Using $\bar{w}$, we calibrate the quantized model with standard layer-wise teacher-forcing on fully observed tokens, minimizing a position-weighted hidden-state MSE. This avoids expensive end-to-end diffusion rollouts during calibration while prioritizing high-impact frontier commits and using masked-stage reliability to obtain a robust, transferable prior.

Crucially, this design stems from the insight that positional vulnerability is an intrinsic structural property dictated by the model weights and decoding dynamics. Therefore, the importance prior estimated from the masked probing phase can be effectively reused in a full-text calibration setting without loss of relevance.

### 3.3. Frontier-Aware Time×Position Weights

We design a position-wise weight $w_i$ by probing the FP teacher decoding dynamics. At each step $t$ along a teacher rollout, we accumulate two additive components in answer region:

$$w_i \leftarrow w_i + \lambda_0(t) \, \mathbf{1}\{i \in \widehat{C}_t\} + \lambda_1 \, \widetilde{c}_{t,i} \, \mathbf{1}\{i \in \mathcal{M}(S_t)\}, \quad (9)$$

where: (i) $\widehat{C}_t$ is the realized write frontier (a sampled $C_t$) in the teacher rollout; $\mathbf{1}\{i \in \widehat{C}_t\}$ is the *frontier-hit* indicator at step $t$. (ii) $\widetilde{c}_{t,i}$ is a row-wise normalized *masked-state teacher sharpness* score (e.g., token-probability, negative entropy, or margin), used as a *reliability gate* when aggregating a transferable static prior for off-policy Stage II calibration. (iii) $\lambda_0(t)$ follows an early-boost schedule to emphasize earlier commits that influence more subsequent steps; $\lambda_1$ scales the fragility term. Importantly, the weight is *additively accumulated* across steps, which will be theoretically justified in Section 4. We probe with random commits to obtain *training-aligned, policy-agnostic* coverage of partially-masked states; all accuracy evaluations still follow each model's default inference-time policy.

**Window alignment and floor.** In prompt-conditioned dLLM generation, diffusion acts primarily on the answer window $\mathcal{G}$ while the prompt is a fixed condition. We therefore align $w$ to the last-$K$ generation window (e.g., $K = 256$) and apply a small floor weight outside $\mathcal{G}$ for numerical stability in layerwise calibration.

### 3.4. Off-Policy Static Teacher-Forcing Calibration with Weighted Hidden MSE

Direct end-to-end optimization over diffusion trajectories during calibration would require rolling out the quantized model for all steps and updating quantization parameters iteratively, which is prohibitively expensive and is incompatible with standard layer-wise PTQ calibration. Instead, we use an off-policy surrogate: rather than calibrating on the on-policy masked states induced by a commit policy, we feed fully observed real tokens (no masks) and align hidden representations between the quantized model and the FP teacher. For each layer/block $\ell$ in a sequential order, we calibrate only $\theta_\ell$ while keeping other layers fixed:

$$\arg\min_{\theta_\ell} \mathbb{E}_{(x,y)\sim\mathcal{D}} \left[ \sum_{i=1}^{N} \bar{w}_i \left\| h_{\ell,i}^q(x,y;\theta_{\leq\ell}) - h_{\ell,i}^\star(x,y) \right\|_2^2 \right], \tag{10}$$

where $\theta_{\leq\ell}$ denotes that layers $< \ell$ have been already calibrated and frozen. Section 4 shows that this objective is a principled surrogate for reducing $\mathrm{KL}(\mu^\star\|\mu^q)$ under mild assumptions, where $\mu^\star$ and $\mu^q$ are the final decoded output distributions of the teacher and the quantized model (formalized in Section 4.1).

## 4. Theoretical Analysis

**Takeaway.** Under model-independent random commits and mild smoothness, $\mathrm{KL}(\mu^\star\|\mu^q)$ admits an additive time×position upper bound with contributions only from committed positions, and each term is controlled by a squared hidden-state discrepancy—justifying our Stage II $\bar{w}$-weighted hidden-state MSE surrogate.

Our analysis proceeds in three steps: (i) upper bound the output KL by a trajectory KL and decompose it across timesteps; (ii) show each per-step divergence only involves committed positions; (iii) bound token-level KL by squared logit error and then by hidden-state MSE, yielding a tractable weighted surrogate. All necessary proofs and additional remarks are provided in Appendix B.

### 4.1. Output Divergence Objective

Let $\tau = (S_T, S_{T-1}, \ldots, S_0)$ denote a decoding trajectory. Under policy $\pi$ and model $M$, the induced trajectory distri-

bution is

$$\mathbb{P}^M(\tau) = p(S_T) \prod_{t=1}^{T} K_t^M(S_{t-1} \mid S_t), \tag{11}$$

where $K_t^M$ is the one-step transition kernel. The output distribution is the marginal of $S_0$:

$$\mu^M(S_0) = \sum_{\tau:S_0(\tau)=S_0} \mathbb{P}^M(\tau). \tag{12}$$

Our ultimate theoretical objective for calibration is distribution alignment between FP and quantized outputs:

$$\min \ \mathrm{KL}(\mu^\star\|\mu^q), \tag{13}$$

which is intractable to compute directly, because evaluating $\mu^M$ requires marginalizing over all possible commit-set choices and token assignments across $T$ steps.

### 4.2. From Output Divergence to Trajectory Divergence

The objective $\mathrm{KL}(\mu^\star\|\mu^q)$ compares the *final* decoded outputs, but it is difficult to evaluate because $\mu^M$ marginalizes over all intermediate commit decisions across $T$ steps. We therefore upper bound the output divergence by the divergence between the *entire decoding trajectories*, which admits a Markovian decomposition into per-step terms.

**Lemma 4.1** (Data processing upper bound). *Let $g(\tau) = S_0$ be the mapping from a trajectory to its final state. Then*

$$\mathrm{KL}(\mu^\star\|\mu^q) \leq \mathrm{KL}(\mathbb{P}^\star(\tau)\|\mathbb{P}^q(\tau)). \tag{14}$$

Lemma 4.1 formalizes that matching the full trajectory distribution is sufficient: any mismatch in outputs must originate from mismatches along the trajectory.

Next, we use the chain rule for KL on Markov path measures, where $d_t^\star$ denotes the teacher's *occupancy measure*, i.e., the marginal distribution of $S_t$ when rolling out the teacher from $p(S_T)$.

**Lemma 4.2** (Markov chain KL decomposition). *Assume both chains share the same initial distribution $p(S_T)$. Let $\mathbb{P}^M(\tau) = p(S_T) \prod_{t=1}^{T} K_t^M(S_{t-1} \mid S_t)$ for $M \in \{\star, q\}$. Then*

$$\begin{aligned} &\mathrm{KL}(\mathbb{P}^\star(\tau)\|\mathbb{P}^q(\tau)) \\ &= \sum_{t=1}^{T} \mathbb{E}_{S_t\sim d_t^\star} \left[ \mathrm{KL}(K_t^\star(\cdot \mid S_t) \| K_t^q(\cdot \mid S_t)) \right], \end{aligned} \tag{15}$$

*where $d_t^\star$ is the teacher occupancy measure at step $t$.*

Together, Lemma 4.1 and Lemma 4.2 reduce the problem to bounding the one-step kernel divergence $\mathrm{KL}(K_t^\star(\cdot \mid S_t)\|K_t^q(\cdot \mid S_t))$ under teacher-visited states. We next show this per-step divergence is *sparse* and only involves the committed positions.

## 4.3. Per-Step KL Decomposition under Commit Policies

To derive a *structural, policy-agnostic* time×position prior, we analyze the Markov kernel divergence under *model-independent random commits*, where it admits an exact sparse decomposition over committed positions. We formalize the intuition that *only committed positions contribute to the per-step KL*.

Let $\pi$ be a commit-set distribution and assume it is model-independent under random commit. Given $S_t$, we sample $C_t \sim \pi(\cdot \mid S_t)$ and then sample tokens for $i \in C_t$ from $p_{M,i}(\cdot \mid S_t)$, while copying all other positions deterministically. The transition kernel can be expressed as:

$$
K_t^M(S_{t-1} \mid S_t)
$$
$$
= \sum_{C_t \subseteq \mathcal{M}(S_t)} \pi(C_t \mid S_t) \left[ \prod_{i \in C_t} p_{M,i}(S_{t-1}[i] \mid S_t) \right] \quad (16)
$$
$$
\cdot \mathbf{1}\{S_{t-1}[\neg C_t] = S_t[\neg C_t]\}.
$$

**Assumption 4.3** (Disjoint-support monotone-mask transitions). For any state $S_t$, committed positions always emit tokens in $\mathcal{V}$ (excluding `[MASK]`), and uncommitted positions deterministically remain `[MASK]`. Equivalently, the mask pattern of $S_{t-1}$ uniquely determines the realized commit set $C_t$ via $\mathcal{M}(S_{t-1}) = \mathcal{M}(S_t) \setminus C_t$. Hence, conditional next-state distributions induced by different commit sets have disjoint support. Notably, the decoding of both Dream and LLaDA is ***monotone by design***: once a position is committed to a vocabulary token, it is never remasked in later steps.

**Proposition 4.4** (Per-step KL reduces to committed positions). *Assume the commit-set distribution $\pi(\cdot \mid S_t)$ is model-independent (random commit) and Assumption 4.3 holds. Then for any fixed $S_t$,*

$$
\mathrm{KL}(K_t^\star(\cdot \mid S_t) \| K_t^q(\cdot \mid S_t))
$$
$$
= \mathbb{E}_{C_t \sim \pi(\cdot \mid S_t)} \left[ \sum_{i \in C_t} \mathrm{KL}(p_{\star,i}(\cdot \mid S_t) \| p_{q,i}(\cdot \mid S_t)) \right].
$$
$$
(17)
$$

Combining Lemmas 4.1–4.2 and Proposition 4.4 yields an upper bound:

$$
\mathrm{KL}(\mu^\star \| \mu^q)
$$
$$
\leq \sum_{t=1}^{T} \mathbb{E}_{S_t \sim d_t^\star} \mathbb{E}_{C_t} \left[ \sum_{i \in C_t} \mathrm{KL}(p_{\star,i}(\cdot \mid S_t) \| p_{q,i}(\cdot \mid S_t)) \right].
$$
$$
(18)
$$

Proposition 4.4 shows that, under model-independent commits, the per-step divergence decomposes into a sum of token-level KL terms *only on committed positions*. This yields a "sum over time, then sum over positions" structure, which directly motivates additive time×position accumulation.

*Remark* 4.5 (Inference-time policies vs. random-commit analysis). Proposition 4.4 assumes a model-independent commit distribution $\pi$ (random commit), which yields an exact sparse decomposition over committed positions and motivates additive time×position accumulation. We intentionally analyze and probe under this *training-aligned* process: random masking/commits match the corruption used in dLLM pretraining/SFT (e.g., independent random masks; cf. LLaDA (Nie et al., 2025)), and provide broad, policy-agnostic coverage of partially-masked states. In practice, inference adopts model-dependent commit policies (Dream: entropy-driven; LLaDA: confidence-driven), so teacher and quantized models may induce different commit sets, introducing an additional policy-shift term (Appendix B.1). Stage I yields a policy-agnostic structural prior, and we verify improvements under each model's default inference-time policy via reduced teacher-forced commit-step flips and suppressed non-teacher-forced error amplification (Section 5.4).

With Proposition 4.4, Eq. (18) makes explicit which $(t, i)$ updates can contribute to the trajectory divergence. We therefore construct a static position prior by (i) Monte Carlo frontier-hit accumulation with time reweighting $\lambda_0(t)$, and (ii) a masked-stage reliability gate to stabilize the estimate for off-policy reuse in Stage II; see Appendix B.9.

## 4.4. From Token KL to Squared Logit Error via Smoothness

Define the log-sum-exp function $f(z) = \log \sum_v \exp(z_v)$. Its gradient is $\nabla f(z) = \mathrm{softmax}(z)$ and its Hessian is

$$
\nabla^2 f(z) = \mathrm{Diag}(p) - pp^\top, \quad p = \mathrm{softmax}(z), \quad (19)
$$

whose operator norm satisfies $\|\nabla^2 f(z)\|_2 \leq \frac{1}{2}$ for all $z$. Hence $f$ is $(1/2)$-smooth. This property holds because the maximum eigenvalue of a covariance matrix of a categorical distribution is at most $(1/2)$.

**Lemma 4.6** (Softmax KL is a Bregman divergence bounded by squared logit error). *Let $p = \mathrm{softmax}(z)$ and $q = \mathrm{softmax}(z')$. Then*

$$
\mathrm{KL}(p \| q) = D_f(z' \| z) \leq \frac{1}{4} \|z' - z\|_2^2, \quad (20)
$$

*where $D_f(u \| v) = f(u) - f(v) - \langle \nabla f(v), u - v \rangle$ is the Bregman divergence of $f$.*

## 4.5. Bridging to Weighted Hidden-State MSE

Define the *suffix network* $g_\ell$ as the mapping from layer-$\ell$ hidden states to logits at the same position (i.e., the remaining blocks, final normalization, and output head), so that $z_{M,i}(S_t) = g_\ell(h_{\ell,i}^M(S_t))$ for $M \in \{\star, q\}$. Assume $g_\ell$ is $L_\ell$-Lipschitz on the calibration domain: $\|g_\ell(u) - g_\ell(v)\|_2 \leq$

*Table 1.* **W4A4 results on the LLaDA family.** Accuracy (%) on PIQA, BoolQ, WinoGrande, ARC-E, ARC-C, HellaSwag, TruthfulQA-MC2, MMLU, HumanEval, and GSM8K. Higher is better.

| Model | Method | PIQA | BoolQ | Wino. | ARC-E | ARC-C | Hella. | Truth | MMLU | Human. | GSM8K | Avg. |
|---|---|---|---|---|---|---|---|---|---|---|---|---|
| LLaDA-Base | FP | 74.84 | 63.73 | 73.64 | 75.08 | 47.44 | 72.90 | 45.30 | 65.80 | 33.54 | 68.92 | 62.12 |
| | RTN | 69.26 | 64.01 | 61.80 | 64.31 | 34.47 | 60.41 | 38.80 | 51.05 | 12.20 | 35.03 | 49.13 |
| | QuaRot | 73.61 | **65.69** | 72.22 | 72.35 | 46.16 | 69.96 | 43.35 | 62.04 | 25.00 | 57.39 | 58.78 |
| | FlatQuant | 74.16 | 62.51 | 72.16 | 73.23 | 46.84 | 71.15 | 42.90 | 63.80 | 29.70 | 57.24 | 59.37 |
| | **FAIR-Calib** | **74.92** | 62.69 | **72.93** | **74.45** | **48.38** | **71.27** | **43.91** | **64.11** | **33.54** | **64.75** | **61.09** |
| LLaDA-Instruct | FP | 82.86 | 88.38 | 77.35 | 94.00 | 88.47 | 76.89 | 48.47 | 64.36 | 46.95 | 70.36 | 73.81 |
| | RTN | 77.26 | 84.68 | 70.09 | 88.71 | 79.66 | 65.93 | 45.17 | 57.27 | 37.80 | 61.71 | 66.83 |
| | QuaRot | 81.23 | 87.98 | 75.06 | **93.30** | 85.42 | 73.27 | **47.77** | 61.99 | 40.24 | 67.10 | 71.34 |
| | FlatQuant | 81.83 | 87.98 | 75.53 | 92.55 | 87.31 | 74.15 | 46.39 | **62.58** | 39.02 | 66.45 | 71.38 |
| | **FAIR-Calib** | **82.10** | **88.29** | **76.01** | 92.77 | **87.46** | **74.50** | 47.44 | **62.58** | **43.29** | **69.60** | **72.40** |
| LLaDA-1.5 | FP | 82.97 | 88.47 | 77.27 | 93.12 | 87.12 | 76.91 | 48.76 | 64.51 | 46.95 | 69.22 | 73.53 |
| | RTN | 76.28 | 85.99 | 70.09 | 88.71 | 82.37 | 66.53 | 46.39 | 57.78 | 38.41 | 61.49 | 67.40 |
| | QuaRot | 80.74 | 87.07 | 75.53 | 91.89 | 85.82 | 73.96 | 47.46 | 61.97 | 42.07 | 63.99 | 71.05 |
| | FlatQuant | 81.28 | 86.92 | **76.01** | 92.95 | 85.12 | 73.97 | 46.48 | 62.96 | 46.34 | 67.40 | 71.94 |
| | **FAIR-Calib** | **82.70** | **87.89** | **76.01** | **94.00** | **86.10** | **74.29** | **48.01** | **63.12** | **46.95** | **68.46** | **72.75** |

*Table 2.* **W4A4 results on the Dream family.** Accuracy (%) on the same benchmark suite. Higher is better.

| Model | Method | PIQA | BoolQ | Wino. | ARC-E | ARC-C | Hella. | Truth | MMLU | Human. | GSM8K | Avg. |
|---|---|---|---|---|---|---|---|---|---|---|---|---|
| Dream-Base | FP | 75.41 | 84.46 | 73.32 | 83.84 | 59.13 | 73.65 | 44.17 | 71.36 | 58.54 | 76.19 | 70.01 |
| | RTN | 61.70 | 59.63 | 55.64 | 51.60 | 34.30 | 55.20 | 41.42 | 36.94 | 6.71 | 16.83 | 42.00 |
| | QuaRot | 72.52 | 74.77 | 63.46 | 75.72 | 48.72 | 66.92 | 40.40 | 59.74 | 23.17 | 49.51 | 57.49 |
| | FlatQuant | 71.65 | 79.14 | 65.98 | 77.19 | 50.51 | 69.54 | 43.23 | 64.96 | 38.41 | 60.20 | 62.08 |
| | **FAIR-Calib** | **73.07** | **80.31** | **69.53** | **80.51** | **53.92** | **70.60** | **43.40** | **66.91** | 41.46 | **66.64** | **64.64** |
| Dream-Instruct | FP | 75.79 | 85.66 | 72.69 | 84.68 | 61.43 | 73.89 | 47.12 | 69.79 | 57.93 | 81.10 | 71.01 |
| | RTN | 63.33 | 59.94 | 56.04 | 59.51 | 40.53 | 56.54 | 41.12 | 41.28 | 13.05 | 26.29 | 45.76 |
| | QuaRot | 71.11 | 76.91 | 64.96 | 78.20 | 52.30 | 67.28 | 39.92 | 64.17 | 32.32 | 64.25 | 61.14 |
| | FlatQuant | 71.82 | 79.66 | 66.38 | 80.89 | 55.29 | 69.82 | 44.27 | **64.18** | 41.46 | 66.03 | 63.98 |
| | **FAIR-Calib** | **73.12** | **81.77** | **70.80** | **83.46** | **58.02** | **70.98** | **46.99** | 64.04 | **44.51** | **72.86** | **66.66** |

$L_\ell \|u - v\|_2$. Then $\|z_{q,i}(S_t) - z_{\star,i}(S_t)\|_2^2 \leq L_\ell^2 \|h_{\ell,i}^q(S_t) - h_{\ell,i}^\star(S_t)\|_2^2$. Combining Lemma 4.6 with Proposition 4.4 yields

$$\text{KL}(\mu^\star \| \mu^q)$$
$$\leq \frac{L_\ell^2}{4} \sum_{t=1}^{T} \mathbb{E}_{S_t \sim d_t^\star} \mathbb{E}_{C_t} \left[ \sum_{i \in C_t} \|h_{\ell,i}^q(S_t) - h_{\ell,i}^\star(S_t)\|_2^2 \right]. \tag{21}$$

Details of the suffix-network Lipschitz bridge are provided in Appendix B.3. This provides a principled justification for minimizing a *weighted hidden-state MSE* as a KL-consistent surrogate, and explains why directly applying softmax-KL to hidden features is unnecessary.

## 5. Experiments

### 5.1. Implementation Details

We evaluate W4A4 on LLaDA and Dream, comparing RTN/QuaRot/FlatQuant under matched PTQ settings. FAIR-Calib probes a fixed prior from the FP teacher and performs weighted layer-wise calibration; details in Appendix C.

### 5.2. Main Results

We report W4A4 accuracy on a broad benchmark suite for LLaDA and Dream, comparing FP, RTN, QuaRot, FlatQuant, and FAIR-Calib under the implementation details described above. Tables 1 and 2 summarize the results for the LLaDA and Dream families, respectively. Overall, FAIR-Calib consistently improves over baselines while using a shorter calibration sequence length (1024) by default.

### 5.3. Ablation Study

**Ablation on components.** Table 3 shows that using either **frontier-hit only** or **masked-stage reliability only** improves over the uniform PTQ baseline, while combining them (**FAIR-Calib**) achieves the best average accuracy on Dream-Base across the 10-benchmark suite. This suggests the two signals are complementary: frontier hits prioritize positions with high downstream impact due to irreversible commits, while masked-stage reliability provides an offline reliability prior that, in expectation, downweights positions where the teacher is frequently ambiguous during masking, thereby reducing the finite-sample noise when estimating and reusing a static prior across corpora in Stage II.

*Table 3.* **Ablation on components.** Average W4A4 accuracy (%) on Dream-Base over the same 10-benchmark suite. **frontier-hit only** keeps only the write-frontier indicator term with $\lambda_0(t)$; **masked-stage reliability only** keeps only the masked-stage reliability term with $\lambda_1$; **FAIR-Calib** combines both.

| Method | Avg. |
| --- | --- |
| baseline | 61.76 |
| frontier-hit only | 63.12 |
| masked-stage reliability only | 62.89 |
| FAIR-Calib | **64.64** |

*Table 4.* **Probing and time-weighting ablations.** Average W4A4 accuracy (%) on Dream-Base over the same 10-benchmark suite. **Left:** varying the probing budget $N_{\text{probe}}$ for estimating the prior $\bar{w}$. **Right:** the frontier-hit time-weight schedule $\lambda_0(t)$.

*(a)* **Probing budget $N_{\text{probe}}$.**

| $N_{probe}$ | Avg. |
| --- | --- |
| 128 | 63.15 |
| 256 | 63.56 |
| 512 | **64.64** |
| 1024 | 64.63 |

*(b)* **Time weighting $\lambda_0(t)$.**

| $\lambda_0$ schedule | Avg. |
| --- | --- |
| w/o $\lambda_0$ | 62.89 |
| uniform | 63.21 |
| late-boost | 62.58 |
| early-boost | **64.64** |

**Sensitivity to probing budget.** Table 4a varies the probing sample size $N_{\text{probe}}$ used in Stage I. Accuracy improves with $N_{\text{probe}}$ and saturates around 512–1024 samples, suggesting that $\bar{w}$ can be estimated reliably with a moderate probing budget. Unless stated otherwise, we use $N_{\text{probe}}$=512.

**Effect of time weighting.** Table 4b compares different $\lambda_0(t)$ schedules for the frontier-hit term. Early-boost performs best, consistent with the intuition that earlier commits influence more subsequent refinement steps and thus have higher downstream impact. Late-boost underperforms, suggesting that emphasizing late commits is less effective at mitigating irreversible error amplification.

### 5.4. Mechanistic Diagnostics

**False-commit error amplification.** We first measure the end-to-end consequence under *inference-time* commit policy. We track the per-step discrepancy w.r.t the FP teacher (probability MSE) and mark the first *false commit* where the written token disagrees with the teacher. Figure 3 shows that, for the uniformly calibrated baseline, a single false commit is typically followed by an abrupt rise in discrepancy that persists over subsequent refinement steps, consistent with irreversibility at the write frontier. In contrast, **FAIR-Calib** substantially **suppresses** this error growth, yielding a flatter discrepancy trajectory after the first wrong write. This motivates a controlled test that isolates whether quantization increases *write-decision flips* at the frontier.

**Teacher-forced commit-step flips.** To isolate write-decision errors from state-distribution shift, we evaluate quantized models on teacher-forced intermediate states

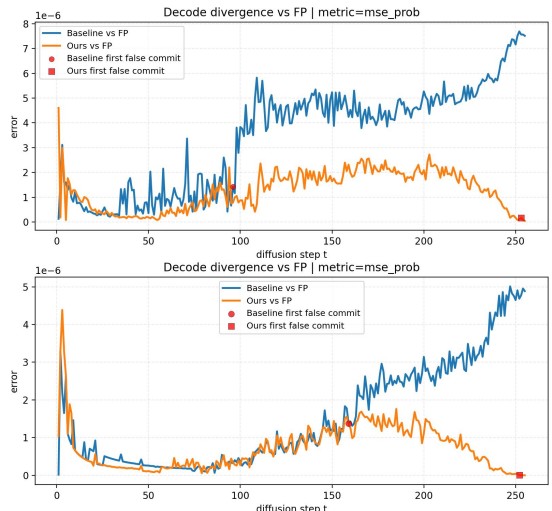

*Figure 3.* Probability MSE w.r.t the FP ($\text{mse\_prob}$) as a function of diffusion step. Wrong commit amplifies downstream error, while FAIR-Calib suppresses this error.

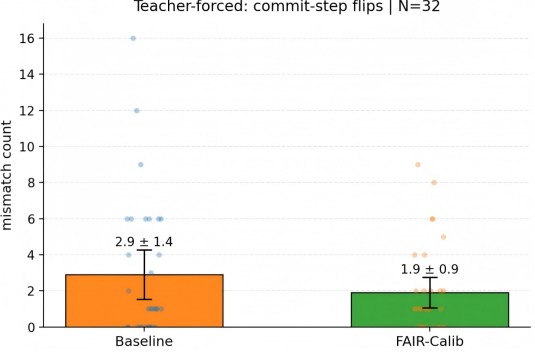

*Figure 4.* **Teacher-forced commit-step flips.** On teacher-forced states, we count write-decision mismatches w.r.t the FP teacher at commit steps (mean±std over N=32). FAIR-Calib reduces flip, showing better alignment at the irreversible write frontier.

along diffusion steps. We compare the token a quantized model would write at the teacher's commit positions against the teacher's write decision, and count mismatches per sequence (a flip at $(t, i)$). FAIR-Calib reduces the average flip count from $2.9 \pm 1.4$ to $1.9 \pm 0.9$ over $N$=32 samples (Figure 4), indicating improved alignment at the write frontier.

**Cross-corpus rank consistency of position weights.** We find that the probed prior $\bar{w}$ is largely *mechanism-driven* rather than corpus-specific: mean normalized $\bar{w}$ curves computed on GSM8K-CoT and WikiText2 exhibit substantial rank consistency (Spearman). Notably, the masked-stage reliability gate is important for transferability, while the frontier-hit-only variant shows near-zero cross-corpus agreement. Detailed diagnostic is provided in Appendix D.2.

# 6. Conclusion

We propose FAIR-Calib to tackle the instability in dLLM quantization caused by irreversible commit decisions. By integrating a frontier-aware reliability prior with off-policy hidden-state alignment, our framework significantly reduces commit-step flips and downstream error propagation. Empirically, FAIR-Calib outperforms existing PTQ methods (e.g., QuaRot, FlatQuant) on Dream and LLaDA at W4A4 precision, providing an efficient and robust solution for compressing diffusion-based language models.

# Acknowledgements

The work was supported by the National Key Research and Development Program of China (No.2023YFC3306401) and National Natural Science Foundation of China 62576018. This research was also supported by Zhejiang Provincial Natural Science Foundation of China under Grant No. LD24F020007, Beijing Natural Science Foundation L244043; The experimental part was supported by the Ministry of Economic Development of the Russian Federation (agreement identifier 000000C313925P3X0002, grant No 139-15-2025-004 dated 17.04.2025).

# Impact Statement

This work studies post-training quantization for diffusion large language models to reduce memory footprint and inference compute, which can lower deployment cost and energy consumption and improve accessibility on resource-constrained hardware. The proposed method does not introduce new model capabilities; it focuses on calibration procedures and diagnostic metrics for quantized inference. As with many compression techniques, quantization may alter model outputs in subtle ways, so we recommend task-relevant evaluation and monitoring before deployment in user-facing settings. Our experiments use standard public benchmarks and do not involve collecting or inferring sensitive personal information. Overall, we expect the primary impact of this work to be improved efficiency and reproducibility for deploying diffusion-style LLMs.

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

# A. Algorithmic Details of FAIR-Calib

For completeness, we provide the full pseudo-code of FAIR-Calib in Algorithm A, including teacher-probed stability weight construction and the subsequent static weighted calibration.

---

**Algorithm A** FAIR-Calib: teacher-probed stability weights + static weighted calibration

---

**Require:** FP teacher $M^\star$, quant model $M^q$, probing set $\mathcal{D}_{\text{probe}}$, calibration set $\mathcal{D}_{\text{cal}}$, steps $T$, gen window $\mathcal{G}$, schedules $\lambda_0(t), \lambda_1$

**Ensure:** Fixed prior $\bar{w}$ and calibrated quant parameters of $M^q$

    **Stage I: compute a global prior $\bar{w}$ by teacher probing**

    **for** sample $(x, y) \sim \mathcal{D}_{\text{probe}}$ **do**

        Run FP teacher rollout with *random commit* to obtain $\{S_t\}_{t=0}^T$, $\{\widehat{C}_t\}$, and $\{\widetilde{c}_{t,i}\}$

        Accumulate $w_i^{(x,y)} \leftarrow \sum_{t=1}^T \lambda_0(t)\mathbf{1}\{i \in \widehat{C}_t\} + \lambda_1 \widetilde{c}_{t,i}\mathbf{1}\{i \in \mathcal{M}(S_t)\}$

        Normalize $w^{(x,y)}$ (e.g., divide by $\max_i w_i^{(x,y)}$) and add to global sum

    **end for**

    Window-align and normalize the aggregated weights to obtain $\bar{w}$

    **Stage II: layer-wise static calibration with fixed $\bar{w}$**

    **for** layer/block $\ell$ in a sequential layer-wise order **do**

        **for** sample $(x, y) \sim \mathcal{D}_{\text{cal}}$ **do**

            Teacher-forcing forward (full real tokens) through $M^\star$ and current $M^q$

            $\mathcal{L}_\ell \leftarrow \sum_{i=1}^N \bar{w}_i \|h_{\ell,i}^q(x,y) - h_{\ell,i}^\star(x,y)\|_2^2$

            Update only quantization/calibration parameters of layer/block $\ell$ by minimizing $\mathcal{L}_\ell$

        **end for**

    **end for**

---

# B. Additional proofs and remarks

### B.1. Remark: model-dependent commit policies

For confidence-driven commit policies (e.g., top-$k$ on confidence), the commit-set distribution $\pi(C_t \mid S_t, M)$ depends on model outputs, and teacher vs quant may induce different $\pi^\star$ and $\pi^q$. In that case, the per-step kernel KL admits an additional policy-shift term:

$$\text{KL}(K_t^\star \| K_t^q) = \text{KL}(\pi^\star(\cdot \mid S_t) \| \pi^q(\cdot \mid S_t)) + \mathbb{E}_{C_t \sim \pi^\star}\left[\sum_{i \in C_t} \text{KL}(p_{\star,i} \| p_{q,i})\right],$$

under Assumption 4.3. Controlling $\text{KL}(\pi^\star \| \pi^q)$ would require on-policy rollouts or reasoning through discrete commit selection, which is typically expensive and not aligned with layer-wise PTQ. We therefore use *model-independent random commits* in Stage I to fix $\pi$ and obtain the clean per-position decomposition in Proposition 4.4, yielding a stable, policy-agnostic importance prior. Random commits also provide broad coverage of partially masked states without coupling the probe to a specific inference policy. Stage II then focuses on reducing the token-level divergence term via the weighted hidden-state MSE surrogate.

### B.2. Remark: why additive $\lambda_0 + \lambda_1$ is natural

The Markov KL decomposition yields a sum over timesteps of per-step divergences; per-step divergences further sum over updated positions. Thus any importance-weighted surrogate has a canonical linear (additive) accumulation form. Multiplicative coupling is not implied by the bound and may introduce unnecessary variance.

### B.3. Remark: suffix-network Lipschitz bridge (details).

For any intermediate layer index $\ell$, we can write the model as a composition of a prefix up to layer $\ell$ and a *suffix network* $g_\ell$ that maps $h_{\ell,i}$ to the final logits at position $i$ (i.e., blocks $\ell{+}1{:}L$, final layer normalization, and the output projection). This

avoids treating intermediate-layer features as "logits" and makes the bridge precise: $z_{M,i}(S_t) = g_\ell(h_{\ell,i}^M(S_t))$. Assuming $g_\ell$ is $L_\ell$-Lipschitz on the teacher-forced calibration domain is standard in PTQ analyses and can be viewed as a local smoothness condition of the remaining network around the in-domain hidden states. Then the token-level KL at committed positions admits the bound $\mathrm{KL}(p_{\star,i}\|p_{q,i}) \leq \frac{1}{4}\|z_{q,i} - z_{\star,i}\|_2^2 \leq \frac{L_\ell^2}{4}\|h_{\ell,i}^q - h_{\ell,i}^\star\|_2^2$, and plugging this into Proposition 4.4 yields Eq. (21). In Stage II, we calibrate blocks layer-wise by directly minimizing a position-weighted hidden-state MSE, which reduces these per-position hidden discrepancies and thus controls an explicit upper bound on the accumulated token divergences on the write frontier.

### B.4. Remark: Off-policy Stage-II as a practical surrogate

The bound above is stated on teacher-visited masked states $S_t \sim d_t^\star$ under the probing policy (random commit), while Stage II calibration uses teacher-forced fully observed tokens for efficiency. Our goal here is to motivate the *additive time×position* structure and a *KL-consistent* feature-space surrogate: reducing hidden/logit discrepancies at high-$\bar{w}$ positions decreases a principled upper bound on token-level divergences on the write frontier, up to a domain-mismatch residual. We empirically validate that this surrogate indeed improves frontier behavior via reduced teacher-forced commit-step flips and suppressed post-commit error amplification (Section 5.4).

### B.5. Proof of Lemma 4.1

*Proof.* By definition, $\mu^M = g_\#\mathbb{P}^M$ is the pushforward measure of $\mathbb{P}^M$ through $g$. KL divergence contracts under measurable mappings (data processing inequality), hence $\mathrm{KL}(g_\#\mathbb{P}^\star\|g_\#\mathbb{P}^q) \leq \mathrm{KL}(\mathbb{P}^\star\|\mathbb{P}^q)$. $\qquad\square$

### B.6. Proof of Lemma 4.2

*Proof.* Using the chain rule for KL on path distributions:

$$\mathrm{KL}(\mathbb{P}^\star\|\mathbb{P}^q) = \mathbb{E}_{\tau\sim\mathbb{P}^\star}\left[\log \frac{p(S_T)\prod_t K_t^\star(S_{t-1} \mid S_t)}{p(S_T)\prod_t K_t^q(S_{t-1} \mid S_t)}\right] = \sum_{t=1}^T \mathbb{E}_{\tau\sim\mathbb{P}^\star}\left[\log \frac{K_t^\star(S_{t-1} \mid S_t)}{K_t^q(S_{t-1} \mid S_t)}\right]. \tag{B.1}$$

Conditioning on $S_t$ under $\mathbb{P}^\star$ yields

$$\mathbb{E}_{\tau\sim\mathbb{P}^\star}\left[\log \frac{K_t^\star(S_{t-1} \mid S_t)}{K_t^q(S_{t-1} \mid S_t)}\right] = \mathbb{E}_{S_t\sim d_t^\star}\left[\mathbb{E}_{S_{t-1}\sim K_t^\star(\cdot|S_t)} \log \frac{K_t^\star(S_{t-1} \mid S_t)}{K_t^q(S_{t-1} \mid S_t)}\right] \tag{B.2}$$

$$= \mathbb{E}_{S_t\sim d_t^\star}\left[\mathrm{KL}(K_t^\star(\cdot \mid S_t)\|K_t^q(\cdot \mid S_t))\right]. \tag{B.3}$$

Summing over $t$ completes the proof. $\qquad\square$

### B.7. Proof of Proposition 4.4

*Proof.* Condition on a commit set $C_t$. Under random commit, both models share the same mixing weight $\pi(C_t \mid S_t)$, and the conditional kernel factorizes over committed positions:

$$K_{t,C_t}^M(S_{t-1} \mid S_t) = \left[\prod_{i\in C_t} p_{M,i}(S_{t-1}[i] \mid S_t)\right] \cdot \mathbf{1}\{S_{t-1}[\neg C_t] = S_t[\neg C_t]\}. \tag{B.4}$$

All uncommitted positions are deterministic copies and thus contribute zero KL, yielding

$$\mathrm{KL}\left(K_{t,C_t}^\star(\cdot \mid S_t) \| K_{t,C_t}^q(\cdot \mid S_t)\right) = \sum_{i\in C_t} \mathrm{KL}(p_{\star,i}(\cdot \mid S_t) \| p_{q,i}(\cdot \mid S_t)). \tag{B.5}$$

Finally, by Assumption 4.3, the supports of $\{K_{t,C_t}^M(\cdot \mid S_t)\}_{C_t}$ are disjoint across different $C_t$. Therefore, the KL between the mixtures equals the mixture of KLs (no log-sum slack), giving the desired equality after taking expectation over $C_t$. $\quad\square$

### B.8. Proof of Lemma 4.6

*Proof.* A standard identity gives $\mathrm{KL}(\mathrm{softmax}(z)\|\mathrm{softmax}(z')) = D_f(z'\|z)$. For an $L$-smooth convex function, $D_f(u\|v) \leq \frac{L}{2}\|u - v\|_2^2$. Here $L = 1/2$, thus $D_f(z'\|z) \leq \frac{1}{4}\|z' - z\|_2^2$. $\qquad\square$

## B.9. Why these weights: reweighting the KL upper bound

The bound (Eq. 18) suggests an additive time×position structure that different positions contribute *unequally* to the output divergence: a position matters only when it is updated, and its impact depends on both when it is updated and how sensitive its token distribution is to perturbations. This motivates estimating *structural importance* over $(t, i)$ and then aggregating it into a static position prior. More broadly, reweighting across diffusion timesteps admits a variational interpretation: reweighted diffusion losses correspond to tighter time-dependent variational bounds and can reduce data–model KL, providing theoretical support for principled (monotone) time reweighting beyond uniform ELBO weighting (Kingma & Gao, 2023; Shi & Titsias, 2025).

**(A) Structural importance via frontier hits.** Under model-independent random commits, $\mathbf{1}\{i \in \widehat{C}_t\}$ is an unbiased indicator of whether position $i$ contributes to the per-step kernel divergence at time $t$. Thus, accumulating

$$w_i^{\text{hit}} \triangleq \sum_{t=1}^{T} \lambda_0(t)\, \mathbf{1}\{i \in \widehat{C}_t\} \tag{B.6}$$

can be viewed as a Monte Carlo estimator of the *structural* time×position contribution suggested by Eq. (18), where $\lambda_0(t)$ implements a monotone time reweighting to reflect downstream amplification of early irreversible writes.

**(B) Transferable prior estimation via masked-stage reliability.** Stage II performs *off-policy* teacher-forcing calibration without masks, so $\bar{w}$ must be a *static* prior that is robust to corpus shift and finite probing budget. We therefore add a masked-stage reliability gate

$$w_i^{\text{rel}} \triangleq \sum_{t=1}^{T} \lambda_1\, \widetilde{c}_{t,i}\, \mathbf{1}\{i \in \mathcal{M}(S_t)\}, \tag{B.7}$$

where $\widetilde{c}_{t,i}$ is high when the teacher distribution is sharp while $i$ is masked. This term is not a structural coefficient of the KL decomposition; rather, it reduces the variance of the probed prior by downweighting intrinsically ambiguous masked contexts, which empirically improves cross-corpus rank consistency of $\bar{w}$ and stabilizes its reuse in Stage II.

**Final weight.** We set $w_i \triangleq w_i^{\text{hit}} + w_i^{\text{rel}}$ and window-align/normalize it to obtain the fixed prior $\bar{w}$ used in Stage II.

## C. Implementation Details

**Models and datasets.** We evaluate W4A4 post-training quantization on two diffusion LLM families: **LLaDA** (Base/Instruct/1.5) (Nie et al., 2025; Zhu et al., 2025) and **Dream** (Base/Instruct) (Ye et al., 2025). Unless stated otherwise, we use each model's default inference-time commit policy for all accuracy evaluations (Dream: entropy-driven; LLaDA: confidence-driven). We report performance on a diverse benchmark suite covering commonsense reasoning and NLU (PIQA (Bisk et al., 2020), BoolQ (Clark et al., 2019), WinoGrande (Sakaguchi et al., 2021), ARC-E/C (Clark et al., 2018), HellaSwag (Zellers et al., 2019)), truthfulness (TruthfulQA-MC2 (Lin et al., 2022)), broad knowledge and multi-task understanding (MMLU (Hendrycks et al., 2020)), code generation (HumanEval (Chen, 2021)), and mathematical reasoning (GSM8K (Cobbe et al., 2021)). Following the official evaluation scripts provided in the released repositories, we adopt the repository-default scoring mode for each benchmark.

**Baselines.** We compare FAIR-Calib against vanilla RTN, QuaRot (Ashkboos et al., 2024), and FlatQuant (Sun et al., 2024) under the same W4A4 PTQ setting. All baselines use 128 calibration sequences from WikiText2 (Merity et al., 2016). QuaRot is calibrated with GPTQ, while FlatQuant and RTN use RTN-style reconstruction. Following the default settings of each method, QuaRot and FlatQuant calibrate with sequence length 2048 for Dream and 4096 for LLaDA; unless stated otherwise, FAIR-Calib uses a shorter calibration length of 1024 for both model families.

**FAIR-Calib specifics.** We adopt per-channel and per-token symmetric quantization for weights and activations, respectively. **Stage I (probing)** runs the FP teacher for *T=256* diffusion steps with *random commit* to estimate a fixed prior $\bar{w}$ over a $K$-token generation window (default $K$=256). We probe on 512 GSM8K questions from the *train* split, formatting each prompt with a standard zero-shot CoT instruction (e.g., "Let's think step by step."). We then compute masked-stage reliability scores $\widetilde{c}_{t,i}$ from token probability (row-wise min–max normalized over masked positions). For the frontier-hit term, we use a polynomial early-boost schedule

$$\lambda_0(t) = \lambda_0 \cdot \max\left(\left(\frac{t-1}{T-1}\right)^{\alpha}, \rho\right), \qquad t = 1, \ldots, T, \tag{C.8}$$

where larger $t$ corresponds to earlier steps; we set $\lambda_0=1.0$, $\alpha=1.5$, and $\rho=0.1$. **Stage II (calibration)** performs standard teacher-forcing layer-wise calibration on WikiText2, minimizing the $\bar{w}$-weighted hidden-state MSE, with $\lambda_1=1.0$.

# D. More Mechanistic Diagnostics

## D.1. Post-commit mismatch diagnostics.

We measure *post-commit self-consistency* along a decoding trajectory. For a position $i$ that is first written at step $t_{\text{write}}(i)$ with token $\hat{x}_i$, we re-evaluate the model's top-1 prediction at the same position $i$ for all later steps while keeping the committed token fixed in the state, and record whether the current top-1 prediction matches $\hat{x}_i$. We report: (i) **mean_disagree_rate**, the average fraction of post-commit steps where the top-1 prediction disagrees with $\hat{x}_i$, averaged over committed positions; (ii) **never_agree_rate**, the fraction of committed positions whose top-1 prediction never matches $\hat{x}_i$ at any subsequent step. Figure A shows that quantization increases post-commit mismatch under a uniform baseline, while FAIR-Calib consistently reduces both metrics, with a larger reduction on `never_agree_rate`.

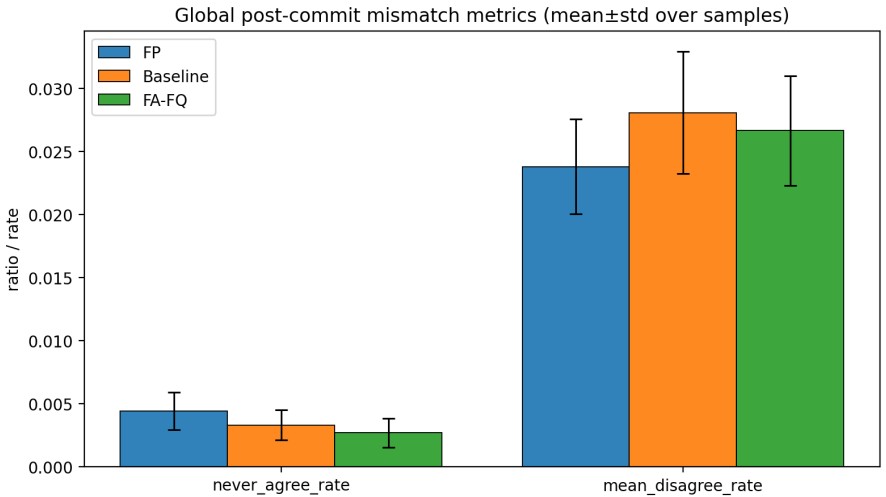

*Figure A.* **Post-commit mismatch metrics.** We report `never_agree_rate` (left) and `mean_disagree_rate` (right), averaged over evaluation samples (mean±std), for the FP teacher, a uniformly calibrated W4A4 baseline, and FAIR-Calib. Quantization exacerbates post-commit mismatch, while FAIR-Calib reduces both metrics, especially the "never-agree" failures.

## D.2. Cross-corpus rank consistency of position weights

To test whether the probed weight prior reflects corpus statistics or decoding dynamics, we compute mean normalized weight curves over the generation window on two corpora (GSM8K-CoT vs. WikiText2) and report Spearman (and Pearson) correlations between the two mean curves. Since per-sample weights are normalized before averaging, the absolute scale of the mean curve (often centered around $\sim 0.5$) is not informative; we focus on the relative ordering across positions (Spearman). Figure B compares four probing regimes:

- (a) *fixed-trajectory random commit* (16 samples; fixed seed), where both corpora share the same commit-set sampling trajectory, yielding high consistency;

- (b) *less-stochastic commits* (256 samples; entropy score with top-$k$ commit), which reduces within-corpus variability but introduces a policy-dependent frontier-update pattern and yields lower cross-corpus agreement;

- (c) *independent-trajectory random commit* (256 samples), where stochastic trajectory variation reduces mean-curve consistency relative to the fixed-trajectory setting, yet substantial correlation remains, indicating a persistent mechanism-driven component.

- (d) Under the same setting as (c), *frontier-hit only* yields near-zero cross-corpus agreement (Spearman $\approx -0.056$), suggesting that frontier-hit statistics alone are dominated by trajectory noise and corpus-specific effects, and motivating the masked-stage reliability gate for constructing a transferable static prior.

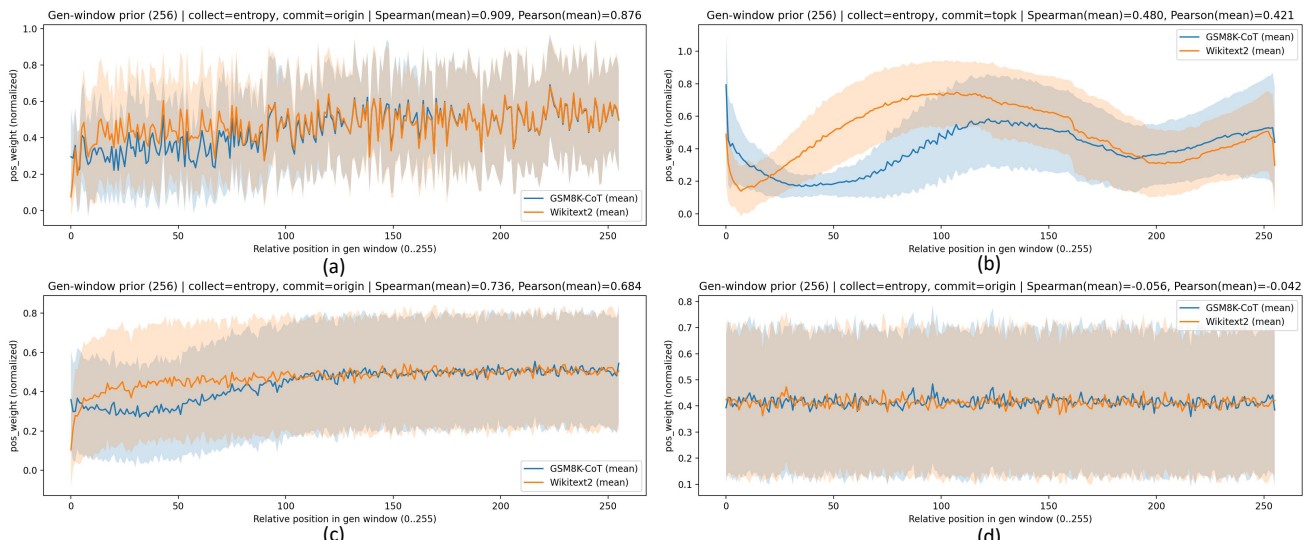

*Figure B.* **Cross-corpus consistency of the time×position weight prior.** Mean normalized weight curves over the 256-token generation window for GSM8K-CoT (blue) and WikiText2 (orange) under four probing settings (top to bottom). Titles report Spearman and Pearson correlations between the two mean curves; shaded bands visualize across-sample variability.

### D.3. Decision-Margin Preservation at Commit Steps

To further substantiate whether FAIR-Calib better preserves write-relevant decisions after quantization, we analyze the frontier decision margin at commit steps. We first run the full-precision teacher to obtain a reference decoding trajectory. For each diffusion step $t$ and position $i$, let $x_{\text{fin}}(i)$ denote the token finally committed at position $i$ by the teacher decoding trajectory. We then evaluate each quantized model on the same pre-commit state from the teacher trajectory and compute its logit margin with respect to this teacher-committed target token:

$$\text{margin}(t, i) = \text{logit}_{t,i}(x_{\text{fin}}(i)) - \max_{v \neq x_{\text{fin}}(i)} \text{logit}_{t,i}(v), \tag{D.9}$$

where the logits are produced by the quantized student model. In other words, the teacher trajectory provides the reference target token $x_{\text{fin}}(i)$, while the margin measures how strongly the quantized model ranks this teacher-committed token over its strongest competing alternative under the same write-frontier state.

As shown in Figure C, FAIR-Calib achieves a higher average frontier margin at commit steps than the unweighted baseline. Since both methods are evaluated against the same teacher-committed target token $x_{\text{fin}}(i)$, this diagnostic directly measures whether quantization changes the model's preference away from the teacher's write decision. The average margin increases from $6.56\pm0.62$ to $7.23\pm0.64$, providing more direct evidence that the proposed frontier-aware weighting better preserves fragile write-relevant decisions after quantization.

### D.4. Stability Lag Analysis for Dream and LLaDA

We further analyze why FAIR-Calib yields larger gains on Dream than on LLaDA. We attribute this difference to their distinct decoding dynamics. In particular, we measure the *stability lag* of commit decisions, which characterizes how long a position remains fragile after it is selected for commitment. A heavier tail in the stability-lag distribution indicates that the model's decisions stay unstable for more steps, making them more vulnerable to quantization-induced perturbations.

As shown in Figure D, Dream exhibits a heavier tail in instability, which means its decisions stay fragile for longer after the commit step compared to LLaDA. Since FAIR-Calib is specifically designed to protect these fragile states, it naturally yields higher gains where the "stability lag" issue is more acute. LLaDA is inherently more stable, yet FAIR-Calib still provides consistent, non-trivial improvements.

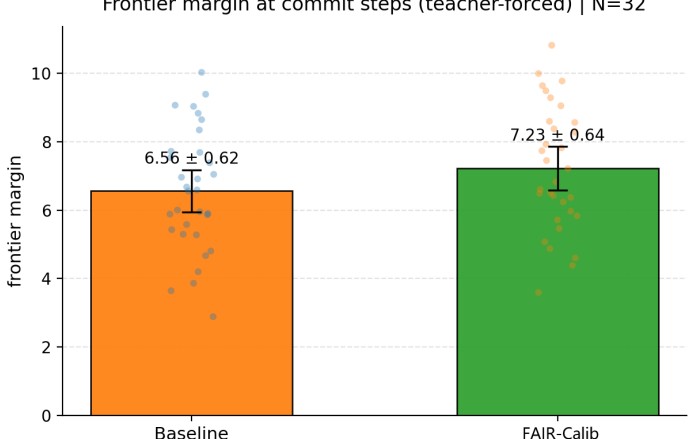

*Figure C.* **Average frontier margin at commit steps.** FAIR-Calib better preserves the decision margin of fragile write-frontier states after quantization. The average frontier margin increases from $6.56\pm0.62$ for the unweighted baseline to $7.23\pm0.64$ under FAIR-Calib.

*Table A.* **Comparison on LLaDA-Instruct under W3A4 quantization.** Compared with FlatQuant, FAIR-Calib improves the overall average from 69.58 to 70.41, yielding a gain of +0.83 points.

| Method | PIQA | BoolQ | WinoGrande | ARC-e | ARC-c | HellaSwag | TruthfulQA | MMLU | HumanEval | GSM8K-CoT | AVG |
|---|---|---|---|---|---|---|---|---|---|---|---|
| Real-valued | 82.86 | 88.38 | 77.35 | 94.00 | 88.47 | 76.89 | 48.47 | 64.36 | 46.95 | 70.36 | 73.81 |
| FlatQuant | 79.54 | **87.65** | 74.06 | **91.71** | 83.05 | 71.12 | 46.15 | 60.60 | 36.59 | 65.34 | 69.58 |
| FAIR-Calib | **79.88** | 87.52 | **74.90** | **91.71** | **85.42** | **71.29** | **47.52** | **60.78** | **38.41** | **66.66** | **70.41** |

# E. Additional Generalization Results

In this section, we provide two additional studies to further examine the generality of the proposed frontier-aware calibration objective. First, we evaluate FAIR-Calib under a more challenging W3A4 quantization setting, where quantization noise is stronger and commitment-flip errors are expected to be more severe. Second, we integrate the proposed position-weighted objective into a QDrop-style calibration framework that directly optimizes quantization parameters, in order to verify that the proposed objective is not tied to a specific PTQ pipeline.

## E.1. Lower-Bit Quantization under W3A4

We first evaluate FAIR-Calib under the more challenging W3A4 setting. We use the same LLaDA-Instruct model, PTQ pipeline, calibration setup, and evaluation protocol as in the main experiments, and only change the quantization precision from W4A4 to W3A4. We compare FAIR-Calib with FlatQuant under this lower-bit setting.

As shown in Table A, FAIR-Calib remains effective under the lower-bit W3A4 setting. Compared with W3A4 FlatQuant, FAIR-Calib improves the overall average from 69.58 to 70.41, with a gain of +0.83 points. The improvements are especially clear on ARC-c, HumanEval, TruthfulQA, GSM8K-CoT, and WinoGrande, where FAIR-Calib improves the corresponding scores by +2.37, +1.82, +1.37, +1.32, and +0.84 points, respectively. These results suggest that the proposed frontier-aware weighting remains beneficial in the more challenging low-bit regime, where quantization-induced perturbations are stronger and write-frontier states are more vulnerable.

## E.2. Extension to QDrop-Style Calibration

We further examine whether the proposed weighting strategy can be applied beyond the specific calibration pipeline used in FAIR-Calib. To this end, we integrate the proposed position-weighted hidden-state reconstruction objective into a QDrop-style calibration framework. Unlike the main FAIR-Calib pipeline, this setting directly optimizes quantization parameters, including scale and zero-point, rather than relying on affine transformation learning. For a fair comparison, we keep the same W8A8 quantization setting, model, calibration data, and evaluation protocol, and only replace the vanilla calibration objective with the proposed position-weighted version.

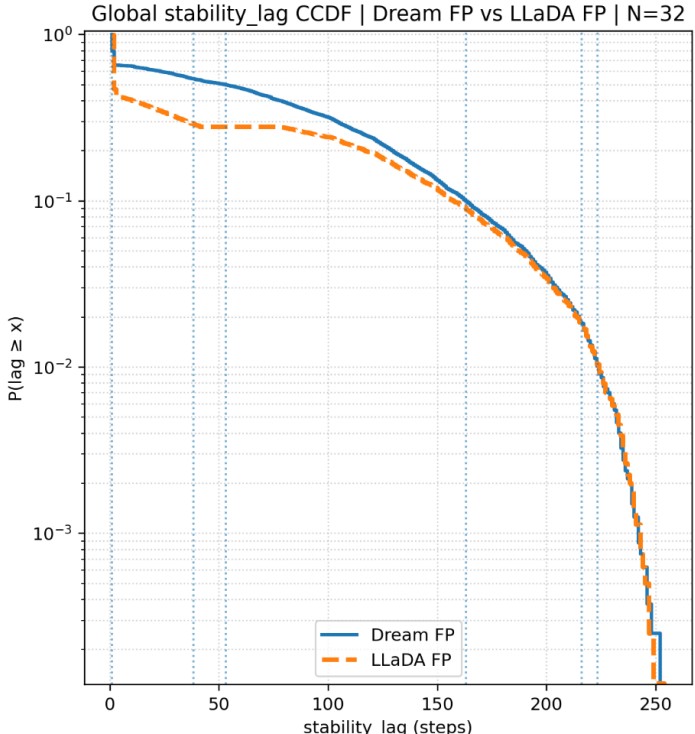

*Figure D.* **Stability-lag analysis for Dream and LLaDA.** Dream exhibits a heavier tail in instability, suggesting that its commit decisions remain fragile for longer after the commit step. This explains why FAIR-Calib tends to yield larger gains on Dream: the proposed frontier-aware objective is designed to protect precisely these fragile write-relevant states.

*Table B.* **Comparison with the QDrop-style baseline under W8A8 quantization.** The proposed position-weighted objective substantially improves the QDrop-style baseline, increasing the average score from 70.66 to 73.17.

| Method | PIQA | BoolQ | WinoGrande | ARC-e | ARC-c | HellaSwag | TruthfulQA | MMLU | HumanEval | GSM8K-CoT | AVG |
|---|---|---|---|---|---|---|---|---|---|---|---|
| Real-valued | 82.86 | 88.38 | 77.35 | 94.00 | 88.47 | 76.89 | 48.47 | 64.36 | 46.95 | 70.36 | 73.81 |
| QDrop-style | 80.69 | 88.20 | 75.45 | 92.06 | 87.46 | 72.21 | 47.14 | 61.78 | 36.59 | 65.05 | 70.66 |
| QDrop-style + position-weighted MSE | **82.93** | **88.44** | **77.19** | **93.30** | **88.47** | 76.74 | 47.14 | 64.06 | 43.29 | 70.13 | 73.17 |
| FAIR-Calib | 82.64 | 88.35 | **77.19** | **93.30** | **88.47** | 76.93 | **48.58** | **64.43** | **46.95** | **71.27** | **73.81** |

As shown in Table B, the proposed position-weighted objective substantially improves the QDrop-style baseline. The average score increases from 70.66 to 73.17, yielding a gain of +2.51 points. This brings the QDrop-style pipeline much closer to the real-valued model, whose average score is 73.81. In addition, the full FAIR-Calib pipeline achieves an average score of 73.81 under W8A8 quantization, matching the real-valued model in overall average performance. These results indicate that the proposed frontier-aware objective is not specific to a single PTQ implementation. Instead, it can serve as a general calibration principle for protecting fragile write-frontier states during post-training quantization.

## F. Efficiency Discussion

We discuss efficiency from two aspects: (i) *calibration efficiency*, and (ii) *compression efficiency*

### F.1. Calibration efficiency

**Stage-I probing cost.** Stage I runs a full-precision teacher for $T$ diffusion steps under random commits, using $N_{\text{probe}}$ samples, and only records lightweight statistics (frontier-hit indicators and masked-stage sharpness) over the $K$-token generation window. Thus the probing complexity is

$$\mathcal{C}_{\text{probe}} = \mathcal{O}(N_{\text{probe}} \cdot T \cdot \text{Cost}(\text{FP forward per step})), \qquad \text{(F.10)}$$

with a small constant factor because no gradients are stored and only a window-aligned weight vector is accumulated. In our default setting ($T=256$, $N_{\text{probe}}=512$, $K=256$), the probing budget is moderate and independent of any layer-wise calibration iterations.

**Stage-II off-policy calibration avoids diffusion rollouts.** A key efficiency feature of FAIR-Calib is that Stage II ***does not*** perform end-to-end diffusion rollouts for the quantized model during calibration. Instead, we use teacher-forcing on fully observed sequences and apply sequential layer-wise reconstruction with the fixed prior $\bar{w}$ (Eq. (10)). This makes the calibration cost comparable to conventional layer-wise PTQ:

$$\mathcal{C}_{\text{cal}} = \mathcal{O}(L \cdot N_{\text{cal}} \cdot \text{Cost(forward/backward on a single layer/block)}), \tag{F.11}$$

where $L$ is the number of blocks and $N_{\text{cal}}$ is the number of calibration sequences (we use 128 sequences from WikiText2 by default). Crucially, this avoids the multiplicative factor of $T$ that would arise if one calibrated by differentiating through diffusion trajectories.

**Shorter sequences and no on-policy state collection.** FAIR-Calib uses a shorter calibration sequence length by default (1024 for both Dream and LLaDA), while some strong baselines calibrate with longer sequences (e.g., 2048/4096 depending on the model family). Moreover, because $\bar{w}$ is computed once in Stage I and reused statically, Stage II does not require collecting on-policy masked states induced by a model-dependent commit policy, which would otherwise increase calibration complexity and engineering overhead.

**Memory footprint during calibration.** Stage II requires comparing teacher and quantized hidden states, but this can be implemented in a streaming manner: teacher activations can be computed under `no_grad` and consumed immediately for the corresponding layer/block calibration. Thus, the peak memory overhead over standard layer-wise PTQ is typically limited to holding (i) the current layer's teacher hidden states and (ii) the quantized layer's activations needed for local reconstruction, rather than storing full diffusion trajectories. The substantially reduced calibration-time memory footprint (Table C) makes FAIR-Calib practical beyond server-class GPUs. In particular, the peak memory during calibration is reduced to $\sim$12–15 GB for LLaDA/Dream in our setup, which falls within the budget of widely available consumer-grade GPUs (e.g., 16–24 GB). This lower footprint allows practitioners to run calibration locally, and also leaves additional headroom to increase the calibration batch size, sequence length, or the number of calibration steps when needed.

*Table C.* **Calibration GPU memory footprint.** Peak GPU memory usage during calibration (reported memory in MB; lower is better). **Reduction** is the ratio `Baseline / FAIR-Calib`.

| Model | Baseline | FAIR-Calib | Reduction |
|---|---|---|---|
| LLaDA-Base | 32923 | 12121 | 2.72× |
| LLaDA-Instruct | 32923 | 12121 | 2.72× |
| LLaDA-1.5 | 32923 | 12121 | 2.72× |
| Dream-Base | 29943 | 14869 | 2.01× |
| Dream-Instruct | 29943 | 14869 | 2.01× |

## F.2. Compression efficiency

**Weight memory reduction under W4A4.** At deployment, low-bit PTQ primarily reduces the storage of model weights. Ignoring small metadata terms, the weight memory scales approximately as

$$\text{Mem(weights)} \approx \frac{b_w}{16} \cdot \text{Mem(FP16 weights)}, \tag{F.12}$$

suggesting an ideal $\sim 4\times$ reduction when $b_w=4$. In practice, the realized memory footprint also includes auxiliary tensors such as per-channel scales, as well as runtime buffers and framework overhead, so the end-to-end reduction is typically smaller than the ideal ratio. Table D summarizes the deployment-time memory footprint after quantization. Across both model families, W4A4 quantization calibrated by FAIR-Calib reduces memory by $\sim$3.1–3.2× compared with FP, while keeping the inference-time decoding procedure unchanged (same number of diffusion steps and commit policy).

*Table D.* **Deployment memory footprint after quantization.** Memory usage in GB for FP vs. W4A4 (FAIR-Calib). **Memory saving** is the ratio `FP / FAIR-Calib`.

| Model | FP (GB) | FAIR-Calib (GB) | Memory saving |
|---|---|---|---|
| LLaDA-Base | 15.89 | 4.91 | 3.24× |
| LLaDA-Instruct | 15.89 | 4.91 | 3.24× |
| LLaDA-1.5 | 15.88 | 4.90 | 3.24× |
| Dream-Base | 13.95 | 4.44 | 3.14× |
| Dream-Instruct | 13.95 | 4.44 | 3.14× |

