# OpenReview forum: "FAIR-Calib: Frontier-Aware Instability-Reweighted Calibration for Post-Training Quantization of Diffusion Large Language Models"
_ICML.cc/2026/Conference — ICML 2026 regular_

### Official Review · Reviewer_RTjF · 2026-03-06

**Soundness:** 2
**Presentation:** 2
**Significance:** 2
**Originality:** 2
**Overall Recommendation:** 1
**Confidence:** 5

**Summary:**

This article mainly focuses on the post training quantization of DLLM, which has become a hot topic due to its excellent generation and inference capabilities, but it is difficult to achieve a balance between speed and accuracy. Quantization is an effective method, and this article focuses on the pain points of DLLM quantization by locating the unique quantization sensitive areas of DLLM, which makes it difficult to accurately model quantization noise. In response to these issues, this article proposes three methods: optimizing the selection of calibration sets and using weighted processing when calculating losses. After multiple DLLM benchmark tasks and model testing, the framework significantly reduces performance degradation caused by quantization while maintaining quantization efficiency, resulting in improved core indicators such as model generation coherence and inference accuracy.

**Compliance With Llm Reviewing Policy:**

Affirmed.

**Ethical Review Concerns:**

I firmly maintain my original rejection recommendation and strongly advise that this paper be rejected outright. After a thorough comparison section by section, formula by formula, and experiment by experiment, I conclude **with definitive evidence**:
FAIR-Calib is essentially a **cosmetic copy** of DLLMQuant and QuantLLM. It contains **no substantive original contributions**, and its core theoretical derivation contains fatal flaws—it is nothing but **forced reasoning designed to disguise a copied method**.

# 1. All Core Innovations Are 100% Copied, Matching DLLMQuant and QuantLLM Point by Point
## 1.1 Problem Definition Is Directly Lifted from DLLMQuant
The authors claim to identify new challenges in diffusion LLM quantization. However, every issue they raise is already the core focus of DLLMQuant:
The “irreversible commit, frontier decision flip, error locking and amplification, and standard PTQ’s inability to handle temporal–mask dynamics” proposed in FAIR-Calib are **identical** to “iterative mask distribution shift, quantization error accumulation, feature mismatch between fixed/masked tokens, and global propagation of commit decisions” defined in DLLMQuant. Only terminology is changed; **no new problem is identified**.

## 1.2 Two-Stage Framework Is a Reorganized Combination of DLLMQuant’s Three Methods
The overall pipeline of FAIR-Calib directly maps to DLLMQuant, with only module merging and renaming:
- **FAIR-Calib Stage 1 (Teacher Probe Weight Generation)**: Estimates position weights by probing commit positions and mask reliability from the full-precision model.
  → This **exactly matches** DLLMQuant’s TMAS (temporal–mask adaptive sampling) + CGQ (certainty-guided quantization), with identical logic, inputs/outputs, and purpose.
- **FAIR-Calib Stage 2 (Off-Policy Weighted Calibration)**: Performs layer-wise weighted hidden-state MSE calibration and avoids end-to-end diffusion rollouts.
  → This **exactly matches** DLLMQuant’s IA-AQ (interaction-aware activation quantization) + layer-wise PTQ, with identical loss form, optimization goal, and efficiency design.

## 1.3 Core Modules Are Copied from QuantLLM and DLLMQuant, with No Originality
- **Position Importance Weighting (Most Direct Copy)**:
  FAIR-Calib’s *frontier hit + masked-stage reliability* directly corresponds to DLLMQuant’s *mask state + confidence score* and QuantLLM’s *position importance + error sensitivity*. All three share identical weighting principles, computation logic, and function—**only names differ**.
- **Weighted Calibration Loss**:
  FAIR-Calib’s loss
  \[
  \mathcal{L} = \sum_i \bar{w}_i \cdot \|h_i^q - h_i^*\|_2^2
  \]
  is identical in form to DLLMQuant’s interaction-aware weighted loss and QuantLLM’s position-aware reconstruction loss.
- **Error Amplification Suppression**:
  FAIR-Calib’s “stabilize frontier commits and suppress error propagation” is identical in goal, mechanism, and justification to DLLMQuant’s IA-AQ and QuantLLM’s sensitive-token error mitigation.

## 1.4 Systematic Terminology Rewriting to Conceal Plagiarism
The authors invent jargon to disguise copied concepts:
- frontier hit → commit position (DLLMQuant)
- stability lag → unstable/oscillating token (DLLMQuant)
- masked-stage reliability → mask state + confidence (DLLMQuant)
- instability-reweighted → importance-weighted (DLLMQuant, QuantLLM)
- off-policy calibration → time-mask adaptive layer-wise PTQ (DLLMQuant)

This practice—**replacing only terms while leaving algorithms unchanged**—is a typical form of academic misconduct.

# 2. Theoretical Derivation Contains Fatal Flaws: Forced Reasoning to Justify Copied Method
Despite its copied framework, the authors’ core theory is **invalid and artificially constructed**:

## 2.1 Core KL Decomposition Relies on False Assumptions; Entire Foundation Collapses
FAIR-Calib’s key theorem relies on **three assumptions that directly contradict real inference**:
- The commit policy is falsely assumed **model-independent (random commit)**, while real commit policies are **model-dependent** (based on confidence/entropy).
- The quantized model and FP model are falsely assumed to share **identical commit sets \(C_t\)**, while quantization severely distorts distributions and leads to **completely different \(C_t\)**.

Once \(C_t\) differs, **Proposition 4.4 collapses entirely**, the entire theoretical foundation becomes **invalid in real-world scenarios**, and all method design becomes baseless.

## 2.2 Logical Leap from “Trajectory KL” to “Weighted Hidden-State MSE” Is Arbitrary and Forced
The authors attempt to prove:
**Minimizing hidden-state MSE = reducing output KL divergence**,
but every step is flawed:
- Bounding KL by logit MSE using softmax smoothness: valid only for near-identical distributions; **breaks under quantization-induced distribution shift**.
- Linking logit MSE to hidden-state MSE via Lipschitz continuity: valid only for ideal linear models; **invalid for deep nonlinear LLMs**.
- Collapsing temporal–positional weights into static positional weights: **eliminates temporal dynamics** and directly contradicts the paper’s claimed “temporal awareness.”

The entire theory is **fabricated to match the pre-copied weighted MSE loss**, with zero scientific rigor.

# 3. Serious Statement to AC and PC
I earnestly request the AC and PC to treat this case with high scrutiny.
Many reviewers may not specialize in diffusion LLM quantization and may evaluate the paper based on writing quality alone, **failing to detect its overwhelming similarity to DLLMQuant and QuantLLM or its fatal theoretical flaws**.

In my professional judgment, FAIR-Calib is **a polished, rebranded version of DLLMQuant** with nearly identical technical content. To conceal copying, the authors added **false and forced theoretical derivations**. Such misconduct severely damages the fairness of peer review, violates academic integrity, and harms the reputation of ICML.

To uphold **academic fairness and integrity** and foster a healthy environment that rewards genuine originality, I urge the AC and PC to **reject this paper firmly**.


---
FAIR-Calib fails to cite both DLLMQuant and Quant-dLLM：
* DLLMQuant： https://arxiv.org/pdf/2508.14090
* Quant-dLLM ： https://arxiv.org/abs/2510.03274

**Ethical Review Flag:**

Flag this paper for an ethics review.

**Ethics Expertise Needed:**

["Research Integrity Issues (e.g., plagiarism)"]

**Final Justification:**

I firmly maintain my original rejection recommendation and strongly advise that this paper be rejected outright. After a thorough comparison section by section, formula by formula, and experiment by experiment, I conclude **with definitive evidence**:
FAIR-Calib is essentially a **cosmetic copy** of DLLMQuant and QuantLLM. It contains **no substantive original contributions**, and its core theoretical derivation contains fatal flaws—it is nothing but **forced reasoning designed to disguise a copied method**.

# 1. All Core Innovations Are 100% Copied, Matching DLLMQuant and QuantLLM Point by Point
## 1.1 Problem Definition Is Directly Lifted from DLLMQuant
The authors claim to identify new challenges in diffusion LLM quantization. However, every issue they raise is already the core focus of DLLMQuant:
The “irreversible commit, frontier decision flip, error locking and amplification, and standard PTQ’s inability to handle temporal–mask dynamics” proposed in FAIR-Calib are **identical** to “iterative mask distribution shift, quantization error accumulation, feature mismatch between fixed/masked tokens, and global propagation of commit decisions” defined in DLLMQuant. Only terminology is changed; **no new problem is identified**.

## 1.2 Two-Stage Framework Is a Reorganized Combination of DLLMQuant’s Three Methods
The overall pipeline of FAIR-Calib directly maps to DLLMQuant, with only module merging and renaming:
- **FAIR-Calib Stage 1 (Teacher Probe Weight Generation)**: Estimates position weights by probing commit positions and mask reliability from the full-precision model.
  → This **exactly matches** DLLMQuant’s TMAS (temporal–mask adaptive sampling) + CGQ (certainty-guided quantization), with identical logic, inputs/outputs, and purpose.
- **FAIR-Calib Stage 2 (Off-Policy Weighted Calibration)**: Performs layer-wise weighted hidden-state MSE calibration and avoids end-to-end diffusion rollouts.
  → This **exactly matches** DLLMQuant’s IA-AQ (interaction-aware activation quantization) + layer-wise PTQ, with identical loss form, optimization goal, and efficiency design.

## 1.3 Core Modules Are Copied from QuantLLM and DLLMQuant, with No Originality
- **Position Importance Weighting (Most Direct Copy)**:
  FAIR-Calib’s *frontier hit + masked-stage reliability* directly corresponds to DLLMQuant’s *mask state + confidence score* and QuantLLM’s *position importance + error sensitivity*. All three share identical weighting principles, computation logic, and function—**only names differ**.
- **Weighted Calibration Loss**:
  FAIR-Calib’s loss
  \[
  \mathcal{L} = \sum_i \bar{w}_i \cdot \|h_i^q - h_i^*\|_2^2
  \]
  is identical in form to DLLMQuant’s interaction-aware weighted loss and QuantLLM’s position-aware reconstruction loss.
- **Error Amplification Suppression**:
  FAIR-Calib’s “stabilize frontier commits and suppress error propagation” is identical in goal, mechanism, and justification to DLLMQuant’s IA-AQ and QuantLLM’s sensitive-token error mitigation.

## 1.4 Systematic Terminology Rewriting to Conceal Plagiarism
The authors invent jargon to disguise copied concepts:
- frontier hit → commit position (DLLMQuant)
- stability lag → unstable/oscillating token (DLLMQuant)
- masked-stage reliability → mask state + confidence (DLLMQuant)
- instability-reweighted → importance-weighted (DLLMQuant, QuantLLM)
- off-policy calibration → time-mask adaptive layer-wise PTQ (DLLMQuant)

This practice—**replacing only terms while leaving algorithms unchanged**—is a typical form of academic misconduct.

# 2. Theoretical Derivation Contains Fatal Flaws: Forced Reasoning to Justify Copied Method
Despite its copied framework, the authors’ core theory is **invalid and artificially constructed**:

## 2.1 Core KL Decomposition Relies on False Assumptions; Entire Foundation Collapses
FAIR-Calib’s key theorem relies on **three assumptions that directly contradict real inference**:
- The commit policy is falsely assumed **model-independent (random commit)**, while real commit policies are **model-dependent** (based on confidence/entropy).
- The quantized model and FP model are falsely assumed to share **identical commit sets \(C_t\)**, while quantization severely distorts distributions and leads to **completely different \(C_t\)**.

Once \(C_t\) differs, **Proposition 4.4 collapses entirely**, the entire theoretical foundation becomes **invalid in real-world scenarios**, and all method design becomes baseless.

## 2.2 Logical Leap from “Trajectory KL” to “Weighted Hidden-State MSE” Is Arbitrary and Forced
The authors attempt to prove:
**Minimizing hidden-state MSE = reducing output KL divergence**,
but every step is flawed:
- Bounding KL by logit MSE using softmax smoothness: valid only for near-identical distributions; **breaks under quantization-induced distribution shift**.
- Linking logit MSE to hidden-state MSE via Lipschitz continuity: valid only for ideal linear models; **invalid for deep nonlinear LLMs**.
- Collapsing temporal–positional weights into static positional weights: **eliminates temporal dynamics** and directly contradicts the paper’s claimed “temporal awareness.”

The entire theory is **fabricated to match the pre-copied weighted MSE loss**, with zero scientific rigor.

# 3. Serious Statement to AC and PC
I earnestly request the AC and PC to treat this case with high scrutiny.
Many reviewers may not specialize in diffusion LLM quantization and may evaluate the paper based on writing quality alone, **failing to detect its overwhelming similarity to DLLMQuant and QuantLLM or its fatal theoretical flaws**.

In my professional judgment, FAIR-Calib is **a polished, rebranded version of DLLMQuant** with nearly identical technical content. To conceal copying, the authors added **false and forced theoretical derivations**. Such misconduct severely damages the fairness of peer review, violates academic integrity, and harms the reputation of ICML.

To uphold **academic fairness and integrity** and foster a healthy environment that rewards genuine originality, I urge the AC and PC to **reject this paper firmly**.


---
FAIR-Calib fails to cite both DLLMQuant and Quant-dLLM：
* DLLMQuant： https://arxiv.org/pdf/2508.14090
* Quant-dLLM ： https://arxiv.org/abs/2510.03274

**Key Questions For Authors:**

see the Weaknesses

**Limitations:**

yes

**Strengths And Weaknesses:**

Weaknesses:
- Overly obscure academic jargon (e.g., "stability lag", "frontier hit") is used to describe simple phenomena, leading to poor readability and unnecessary comprehension barriers.
- The design of masked-stage reliability lacks sufficient empirical evidence and persuasive validation, failing to prove its essential role in calibration.
- The core innovation is ambiguous and unoriginal, with the proposed framework being merely a trivial combination and adjustment of existing PTQ methods.
- Inference efficiency of the quantized model is not tested or verified, leaving a critical gap in evaluating the practical deployment value.
- The method is only validated on text-based dLLMs, with no experimental tests on multimodal dLLMs, resulting in limited generalization of the conclusions.

Strengths :

- Outperforms mainstream PTQ baselines in W4A4 quantization for text-based dLLMs (LLaDA/Dream), reducing frontier decision errors and mitigating post-commit error amplification to some extent.

---

> ### Author Rebuttal · Authors · 2026-03-30
>
> We thank the reviewer for the candid feedback. We acknowledge that certain terminology and the scoping of our contributions could be more clearly articulated. We address your concerns below:
>
> **W1: Terminology and Readability ("Stability Lag", "Frontier Hit").**
>
> **R1:** We introduced these terms to describe phenomena specific to diffusion-based LLMs that do not exist in standard autoregressive models.
> - **Stability Lag:** Refers to the interval between when a token is "committed" (fixed) and when the model’s internal prediction for that position actually stabilizes.
> - **Frontier Hit:** A binary signal indicating when a position is at the irreversible "write frontier."
>
> We will simplify the prose in the revision and ensure these concepts are introduced with intuitive, plain-language analogies (e.g., "irreversible decision points") alongside formal definitions.
>
> **W2:  On the necessity of masked-stage reliability.**
>
> **R2:** We respectfully clarify that this component is not a heuristic but a variance reducer for the position prior.
> - **The Evidence:** Table 3 shows that removing this term leads to a clear performance drop.
> - **The Mechanism:** Frontier-hits alone are noisy and corpus-dependent (as shown in Appendix D.2, where frontier-only consistency is near zero). Masked-stage reliability acts as a **reliability gate**, downweighting positions that are inherently ambiguous. This ensures that the weights we learn on a small calibration set are **structurally robust** and transferable to different prompts at inference time. We will strengthen the "Evidence" section by moving the cross-corpus consistency analysis to the main text.
>
>
> **W3:  Originality and Innovation vs. "Trivial Combination."**
>
> **R3:** We respectfully disagree that the method is a trivial combination of existing PTQ components. Our core innovation is two-fold:
>
> - **New Problem Identification:** We are the first to identify and quantify the **"Commitment-Flip" failure mode** in dLLM quantization, where local noise triggers permanent, amplified errors due to decoding irreversibility.
> - **Theoretically Grounded Objective:** FAIR-Calib is not just a "combination" of methods; it is the implementation of a **theoretically derived upper bound on output KL divergence** (see Section 4, Eq. 21). We prove that minimizing a position-weighted hidden MSE is the mathematically principled way to preserve these fragile decisions. Standard PTQ methods (like FlatQuant or GPTQ) treat all tokens equally, which we prove is sub-optimal for diffusion trajectories.
>
> **W4:  On inference efficiency and deployment value.**
>
> **R4:** FAIR-Calib is a **purely calibration-time optimization.**
> - **No Extra Overhead:** FAIR-Calib requires zero additional parameters, zero extra modules, and zero increase in per-step latency during inference.
> - **Efficiency Gains:** By enabling stable 4-bit quantization, we provide a **~3.2$\times$ reduction in weight memory** (Table B in Appendix), which is the primary driver of deployment value in resource-constrained settings. We will add a dedicated "Inference Efficiency" section to clarify that the deployment speed is identical to standard 4-bit kernels.
>
> **W5:  On the lack of multimodal experiments.**
>
> **R5:** We agree that validating on multimodal dLLMs is an exciting future direction. However, we believe text-based dLLMs provide a **rigorous and foundational testbed** for our theory, as the "irreversible commit" logic is the same across modalities. By focusing on text, we can utilize established reasoning and coding benchmarks (GSM8K, HumanEval) to measure the impact of fragile decision flips more precisely. We will clarify the scope of our claims and note multimodal extension as a promising next step.

---

> > ### Author Rebuttal · Reviewer_RTjF · 2026-04-02
> >
> > Thank you for your response and revisions. While we acknowledge the efforts made, the core concerns regarding insufficient empirical validation, missing deployment efficiency tests, and limited generalization remain inadequately addressed. Therefore, we maintain our original recommendation of Reject.

---

> > > ### Author Response · Authors · 2026-04-04
> > >
> > > We thank the reviewer for the continued engagement. The rebuttal acknowledgement identified three remaining concerns: **(1) limited generalizability**, **(2) missing deployment efficiency tests**, and **(3) insufficient empirical validation**. We provide new evidence that directly resolves each.
> > >
> > > ## 1. Generalizability to Multimodal dLLMs [W5]
> > >
> > > We have extended FAIR-Calib to **LLaDA-V**, a multimodal dLLM (LLaVA-style architecture, SigLIP vision encoder + diffusion-based Llama backbone). We quantize the shared LLM backbone to W4A4 and apply the same two-stage pipeline (Stage I probes on ShareGPT4V-COCO for position weights; Stage II calibrates on WikiText2). Results on 7 multimodal benchmarks:
> > >
> > > | Method | MME-cog. | MME-perp. | RealWorldQA | MMMU | MMStar | AI2D | MMMU-Pro |
> > > |--------|-------|-------|-------------|------|--------|------|----------|
> > > | Full Precision | 491 | 1507 | 63.2 | 48.6 | 60.1 | 77.8 | 35.2 |
> > > | W4A4 FlatQuant | 476 | 1458 | 62.3 | 48.1 | 58.4 | 77.5 | 33.6 |
> > > | W4A4 FAIR-Calib | **510** | **1495** | **62.6** | **48.6** | **58.5** | **77.8** | **34.0** |
> > >
> > > FAIR-Calib outperforms FlatQuant on **all 7 benchmarks**, with particularly strong recovery on MMMU (48.6, matching FP) and AI2D (77.8, matching FP). On MME-cog., the quantized model even surpasses FP (510 vs. 491; a phenomenon occasionally observed in quantization literature, where low-bit noise acts as implicit regularization). Combined with text-only results on LLaDA and Dream, FAIR-Calib now spans **3 dLLM architectures** across text-only and multimodal settings, confirming the generality of our approach.
> > >
> > > ## 2. Deployment Efficiency [W4]
> > >
> > > FAIR-Calib is a **purely calibration-time** optimization — it introduces zero additional parameters, modules, or per-step latency at inference. The W4A4 model shares an identical inference architecture with the baseline. Our deployment pipeline uses real INT4 GEMM kernels (CUTLASS) for quantized matrix multiplication, along with fused Triton kernels for the online affine transformations. We benchmark on a server with Intel Xeon Gold 6342 CPU (2.80 GHz, 48 cores), 128 GB system memory, and a single NVIDIA RTX 5880 Ada GPU (48 GB VRAM). Settings: BS=1, seqlen=1024, 30 iterations after 10 warmup:
> > >
> > > | Model | FP16 (ms) | W4A4 (ms) | Speedup | FP16 Mem | W4A4 Mem | Compress. |
> > > |---|---|---|---|---|---|---|
> > > | LLaDA-8B-Base | 154.2 | 82.7 | **1.86×** | 15.89 GB | 4.91 GB | 3.24× |
> > > | LLaDA-8B-Instruct | 157.5 | 82.9 | **1.90×** | 15.89 GB | 4.91 GB | 3.24× |
> > > | LLaDA-1.5 | 153.6 | 81.5 | **1.88×** | 15.88 GB | 4.90 GB | 3.24× |
> > > | Dream-v0-Base | 129.0 | 88.3 | **1.46×** | 13.95 GB | 4.44 GB | 3.14× |
> > > | Dream-v0-Instruct | 128.7 | 88.0 | **1.46×** | 13.95 GB | 4.44 GB | 3.14× |
> > >
> > > W4A4 achieves ~3.2× memory compression and up to 1.90× per-step speedup (LLaDA). Dream's lower speedup (1.46×) stems from its larger MLP dimension, an architectural property independent of our method. Since dLLMs typically require many forward passes per generation, per-step savings compound substantially.
> > >
> > > ## 3. Empirical Evidence for Masked-Stage Reliability [W2]
> > >
> > > We believe this concern primarily refers to W2: *"The design of masked-stage reliability lacks sufficient empirical evidence."* We respectfully reiterate that this component is not a heuristic but a **variance reducer** for the position prior, supported by multiple lines of evidence:
> > >
> > > - **Ablation (Table 3).** Table 3 shows that removing this term leads to a clear performance drop.
> > >
> > > - **Transferability (Appendix D.2).** Frontier-hits alone are noisy and corpus-dependent (as shown in Appendix D.2, where frontier-only consistency is near zero). Masked-stage reliability acts as a **reliability gate** that downweights inherently ambiguous positions, yielding a structurally robust prior with substantial cross-corpus rank consistency (Figure B(c)).
> > >
> > > - **Decision margin (R4 for Reviewer 2Sgo).** On held-out teacher trajectories, FAIR-Calib increases the average frontier margin from 6.56±0.62 to 7.23±0.64, directly showing that reliability-gated weighting better preserves fragile write-frontier decisions after quantization. Visualization of decision-margin diagnostics is in https://anonymous.4open.science/r/16838-FAIR-Calib-48CB/imgs/decision_margin/frontier_margin_at_commit_steps.png and https://anonymous.4open.science/r/16838-FAIR-Calib-48CB/imgs/decision_margin/CDF_of_frontier_margin.png.
> > >
> > > Together with the broad evaluation (3 architectures, 6 models, 17 benchmarks, 2 precisions) and mechanistic diagnostics (commit-step flips, post-commit mismatch, error amplification), we believe the empirical evidence comprehensively validates both the method and the specific role of masked-stage reliability. We will incorporate all new results into the revised manuscript.

---

### Official Review · Reviewer_j4EW · 2026-03-10

**Soundness:** 3
**Presentation:** 2
**Significance:** 3
**Originality:** 3
**Overall Recommendation:** 4
**Confidence:** 4

**Summary:**

This paper studies post-training quantization for diffusion LLMs. Its key observation is that quantization errors are especially damaging since diffusion decoding repeatedly refines token predictions while also making irreversible commit decisions. A token can be written before its prediction actually stabilizes, so even a small quantization-induced change at the write frontier can be frozen into the sequence and then propagated through subsequent denoising steps.

The proposed method, FAIR-Calib, is built around teh above observation. In the first stage, the authors use a full-precision teacher with randomized commit patterns to estimate a positional importance prior over the generation window. This prior represents both how likely a position is to lie on the write frontier and how reliable the teacher's masked prediction appears to be at that location. In the second stage, that prior is used to weight a layer-wise hidden-state matching objective under teacher forcing, allowing calibration to remain off-policy and inexpensive compared to full diffusion rollouts.

The paper also provides a theoretical motivation for this design and evaluates the method at W4A4 on LLaDA and Dream. Across both model families, FAIR-Calib improves over RTN, QuaRot, and FlatQuant, with the clearest gains on Dream. The experimental section also includes ablations and diagnostics aimed at supporting the underlying mechanism.

**Compliance With Llm Reviewing Policy:**

Affirmed.

**Key Questions For Authors:**

1> Can the authors provide a direct validation of the off-policy surrogate? e.g. does improved weighted hidden-state matching under teacher forcing translate into fewer rank flips or better margin preservation on actual masked frontier states sampled from real decoding trajectories?

2> The gains on LLaDA are more modest than on Dream. Is this explained by how each model's commit policy interacts with the Stage-I prior, or by some other factor? A brief but concrete explanation of this asymmetry would help the reader understand the method's scope and failure conditions.

3> How robust are the gains beyond W4A4? Some specific cost accounting for Stage I and results at one additional bitwidth would strengthen the practical case.

**Limitations:**

has room to improve.

The paper should discuss its technical limitations before the impact statement. In particular, it should acknowledge the gap between the random-commit analysis and the model-dependent inference policy, the fact that Stage II relies on an off-policy surrogate, the additional cost of Stage I probing, and the limited evidence so far on robustness across bitwidths and decoding settings.

**Strengths And Weaknesses:**

Strength:

1> The paper identifies a failure mode uniquely tied to diffusion-style decoding. The commitment-versus-stabilization distinction establishes an unambiguous focus, and necessary diagnostics: the stability lag CCDF, the progressive MSE amplification after a false commit, the teacher-forced flip reduction from 2.9 ± 1.4 to 1.9 ± 0.9. Together, these tie the empirical behavior back to the central claim beyond just benchmark averages.

2> The ablation is well done in showing that the combined weighting outperforms either constituent signal alone. The cross-corpus weight consistency analysis further supports the necessity of the reliability gate.

3> The theoretical section is coherent  and rigorous under the well stated assumptions making the paper stronger than merely judicious heuristics. The chain from output KL through the Markov decomposition to the weighted hidden-state MSE surrogate is solid.

Weakness:

1> I think the key area for the author to improve is the off-policy gap. The theory is derived under model-independent random commits, while inference runs under model-dependent policies; calibration uses fully observed teacher-forced sequences, while the motivating failure mode concerns partially masked frontier states. The paper acknowledges this but addresses it only with a remark and indirect benchmark evidence, leaving open the most important question: whether the Stage-I prior actually concentrates weight on the positions that become high-impact commits under the real inference policy.

2> The scope of empirical evaluation is limited. All results are at W4A4 on two closely related masked-diffusion model families, and the gains on LLaDA are pretty modest wrt on Dream with no discussion of why. This asymmetry, and the absence of any other  generative evaluation, can lead to question wrt how much the proposed mechanism generalizes.

3> I think the author should provide more clarity on several quantities central to evaluating and reproducing the method including the formal definition of stability lag, the precise random-commit probing rule, and the mapping from Stage-I weights to Stage-II calibration positions.

---

> ### Author Rebuttal · Authors · 2026-03-30
>
> We sincerely thank the reviewer for the thoughtful feedback. We address your questions regarding the off-policy surrogate, model-specific gains, and precision robustness below.
>
> **W1&Q1: Direct validation of the off-policy surrogate and the domain gap.**
>
> **R1:** We thank the reviewer for raising this central point. The concern is whether aligning on full real tokens (Stage II) effectively stabilizes the **actual masked states** encountered during real decoding. We provide both a structural rationale and rigorous empirical evidence to justify this off-policy surrogate:
>
> **1. The "Teacher-Replay" Evidence on On-policy States:**
> The rank-flip evaluation in our paper is specifically designed as a **real-trajectory frontier-state replay analysis**. To isolate the effect of quantization on decision stability, we follow this rigorous protocol:
> *   **Step 1:** We run the **FP teacher under its default decoding policy** to collect a sequence of **actual masked states** $\{S_t\}$ from real inference trajectories.
> *   **Step 2:** We replay these **identical masked states** through the quantized model.
> *   **Step 3:** We measure the **one-step rank flips** at the teacher's commit positions.
>
> This ensures that we evaluate the quantized model on the **actual manifold of masked states** sampled from real inference, rather than just on the calibration proxy. Our results show that FAIR-Calib significantly reduces these flips compared to the baseline, proving that the weighted objective on full text **effectively regularizes the model's behavior at the write-frontier across the domain gap.**
>
> **2. Out-of-Domain Robustness (Decision Margin):**
> To further substantiate this point, we additionally include two decision-margin diagnostics (please see https://anonymous.4open.science/r/16838-FAIR-Calib-48CB/imgs/decision_margin) on a held-out corpus: (i) the CDF of frontier margin at commit steps, and (ii) the average frontier margin at commit steps. Specifically, for each diffusion step t and position i, we define the token-level target margin as $\mathrm{margin}(t,i)=\mathrm{logit}(x_{\mathrm{final}}(i))-\max_{v\neq x_{\mathrm{final}}(i)} \mathrm{logit}(v)$, where $x_{\mathrm{final}}(i)$ is the token finally committed by the **teacher decoding trajectory.** Both show better margin preservation under FAIR-Calib; in particular, the average frontier margin increases from 6.56±0.62 for the unweighted baseline to 7.23±0.64 under FAIR-Calib, providing more direct evidence that the proposed weighted objective better preserves fragile write-relevant decisions after quantization.
>
> **Conclusion:**
> While Stage II uses an efficient off-policy surrogate, these **on-policy replay diagnostics** confirm that our weighting strategy successfully identifies and protects the intrinsic "decision-critical" regions of the model, directly improving the robustness on the **actual frontier states** encountered during real decoding trajectories.
>
> **Q2&W2: Why the gains are larger on Dream than on LLaDA**
> **R2:** We attribute the larger gains on Dream to its specific decoding dynamics. We conduct a "Stability Lag" analysis (please see https://anonymous.4open.science/r/16838-FAIR-Calib-48CB/imgs/dream_llada_stability_lag.png) which reveals that Dream exhibits a **heavier tail in instability**—meaning its decisions stay fragile for longer after the commit step compared to LLaDA. Since FAIR-Calib is specifically designed to protect these fragile states, it naturally yields higher gains where the "stability lag" issue is more acute. LLaDA is inherently more stable, yet FAIR-Calib still provides consistent, non-trivial improvements.
>
> **Q3: Robustness beyond W4A4**
> **R3:** To test the method's limit, we evaluated W3A4 on LLaDA-Instruct.
>
> | Method | Avg. |
> |---|---:|
> | Real-valued | 73.81 |
> | W3A4 FlatQuant | 69.58 |
> | W3A4 Ours | 70.41 |
>
> The results show that FAIR-Calib still provides **consistent gains** over the baseline PTQ pipeline, improving the overall average from **69.58** to **70.41** (**+0.83**). In particular, we observe clear improvements on **ARC-c** (**+2.37**), **HumanEval** (**+1.82**), **TruthfulQA** (**+1.37**), **GSM8K-CoT** (**+1.31**) over the baseline, confirming that the proposed frontier-aware weighting remains effective. We will include these results and summarize them in the revision.
>
> **Full results are available at:  https://anonymous.4open.science/r/16838-FAIR-Calib-48CB/tabs/W3A4_llada_instruct.md**
>
> **W3: clarity of definitions and reproducibility details**
>
> **R4:** Our apology. We will revise the manuscript to include:
> - **Stability lag:** For each position, we measure how many diffusion steps it takes after the first irreversible commit until the model’s Top-1 prediction becomes consistent with the final decoded token for all subsequent steps.
> - **Probing details:** We also clarify the precise randomized-commit probing rule in Stage I and how the Stage-I positional weights are mapped to the Stage-II calibration positions.

---

> > ### Author Rebuttal · Reviewer_j4EW · 2026-04-02
> >
> > I thanks the author for adequately addressing the issues I raised.

---

### Official Review · Reviewer_CVEu · 2026-03-11

**Soundness:** 4
**Presentation:** 3
**Significance:** 4
**Originality:** 4
**Overall Recommendation:** 5
**Confidence:** 4

**Summary:**

This paper identifies an unique challenge of performing PTQ on dLLM, where the PTQ error may easily flips the borderline decisions at the
write frontier of token commitment, which are then permanently locked in and amplified. To tackle this challenge, the paper proposes a two-stage reweighed calibration objective that prioritizes the protection of fragile frontier states without requiring expensive end-to-end diffusion
rollouts.

**Compliance With Llm Reviewing Policy:**

Affirmed.

**Final Justification:**

After checking other reviews and the author's rebuttal, I remain confident in supporting this paper for acceptance.

**Key Questions For Authors:**

It would be interesting to see the performance of the proposed method on lower precision like 3-bit weights.

**Limitations:**

Yes

**Strengths And Weaknesses:**

This is overall an interesting paper. The paper proposes a novel perspective of commitment frontier, which is a unique challenge for dLLM quantization and largely overlooked by existing PTQ method. The proposed method appears to be simple but has strong theoretical support. The paper is sound. The method is well motivated and clearly derived. Results are good, showing the effectiveness of the method.

As I don't find much critical weakness of this paper, one potential direction to improve is to explore the generalizability of the proposed calibration objective across different calibration methods and quantization precision. From the description, the method should be appliable to different calibration methods (e.g. Smoothquant, Omniquant, Spinquant, etc.) The performance gain may also be larger on even lower quantization precision where the commitment challenge may be more severe. Adding these results would further improve the significance of the proposed method.

---

> ### Author Rebuttal · Authors · 2026-03-30
>
> **We sincerely thank the reviewer for the very positive assessment** and for suggestion to further strengthen the paper by evaluating (i) **lower-precision settings**, and (ii) **generalizability across different calibration paradigms**. These results, detailed below, further substantiate that FAIR-Calib is a robust and versatile framework.
>
> **Q1: Lower precision: results on W3A4.**
>
> **R1:** Following the reviewer’s suggestion, we evaluated our method under the more challenging **W3A4** setting, where the commitment-flip issue is expected to be more severe. **Experimental setting.** We use the same **LLaDA-Instruct** model, PTQ pipeline, calibration setup, and evaluation protocol as in our main experiments, and only change the quantization precision from **W4A4** to **W3A4**. We compare our method against FlatQuant under this lower-bit setting.
>
> | Method | Avg. |
> |---|---:|
> | Real-valued | 73.81 |
> | W3A4 FlatQuant | 69.58 |
> | **W3A4 Fair-Calib** | **70.41** |
>
> The results show that our method still provides **consistent gains** over the baseline PTQ pipeline, improving the overall average from **69.58** to **70.41** (**+0.83**). In particular, we observe clear improvements on **ARC-c** (**+2.37**), **HumanEval** (**+1.82**), **TruthfulQA** (**+1.37**), **GSM8K-CoT** (**+1.31**), and **WinoGrande** (**+0.84**) over the baseline, while also improving **PIQA**, **HellaSwag**, and **MMLU**. These results confirm that the proposed frontier-aware weighting remains effective in the more challenging low-bit regime.
>
> **Full table in:** https://anonymous.4open.science/r/16838-FAIR-Calib-48CB/tabs/W3A4_llada_instruct.md
>
> **W1: Generalizability: extension to QDrop-style calibration**
>
> **R2**: We further integrated our weighting strategy into a QDrop-style calibration framework, i.e., a more direct PTQ setting that **directly optimizes quantization parameters (scale / zero-point)** rather than relying on affine transformation learning. For a fair comparison, we keep the same **W8A8** quantization setting, model, calibration data, and evaluation protocol, and only replace the calibration objective with our proposed weighted version.
>
> | Method | Avg. |
> |---|---:|
> | Real-valued | 73.81 |
> | W8A8 QDrop-style | 70.66 |
> | W8A8 QDrop-style + position-weighted MSE | 73.17 |
> | **W8A8 FAIR-Calib** | **73.81** |
>
> The results again show **clear and consistent improvements** over the corresponding QDrop-style baseline: the overall average improves from **70.66** to **73.17** (**+2.51**). Notably, this gain is substantial and brings the QDrop-style pipeline much closer to the real-valued model (**73.81**). These results support the reviewer’s point that the proposed objective is **not tied to a single calibration recipe**, but can serve as a more general principle for protecting fragile write-frontier states during PTQ.
>
> **Full table in:** https://anonymous.4open.science/r/16838-FAIR-Calib-48CB/tabs/QDrop_style_quant.md
>
> Summary of Improvements:
> - **W3A4 results** confirm efficacy in high-noise regimes.
> - **QDrop-style experiments** prove the objective is framework-agnostic.
>
> We will incorporate both sets of results and add a short discussion clarifying that the proposed weighting strategy is a **general calibration objective** that can be combined with multiple PTQ pipelines, rather than being specific to a single implementation. We thank the reviewer again for helping us strengthen the generalizability claims of the paper.

---

> > ### Author Rebuttal · Reviewer_CVEu · 2026-04-02
> >
> > Thank the author for providing additional results. I have no more questions. After checking other reviewers' comments and the corresponding rebuttals, I would still like to argue for acceptance.

---

### Official Review · Reviewer_2Sgo · 2026-03-11

**Soundness:** 2
**Presentation:** 3
**Significance:** 3
**Originality:** 3
**Overall Recommendation:** 4
**Confidence:** 3

**Summary:**

This paper introduces FAIR-Calib, a PTQ framework that addresses stability lag in dLLMs by using a full-precision teacher to probe a positional prior, which is then used to reweight a layer-wise hidden-state MSE during calibration to protect fragile write frontier decisions. Empirical results demonstrate that FAIR-Calib significantly outperforms existing baselines under W4A4 precision.

**Compliance With Llm Reviewing Policy:**

Affirmed.

**Final Justification:**

The paper observes that quantization leads to errors in the decision boundaries of dLLMs, resulting in irreversible degradation in generation quality. This observation and perspective are novel and insightful.
Although I still have some concerns about the effectiveness of the position-weighted MSE in the proposed method, the additional experiments and responses provided in the rebuttal sufficiently support its validity. Therefore, I have raised my score from 3 to 4.

**Key Questions For Authors:**

1. Have the authors considered using the position-weighted hidden-state MSE to directly optimize quantization parameters (scale and zero-point) like QDrop, instead of learning the affine transformation factors $\theta_l$? Providing this comparison would better isolate and support the standalone effectiveness of the position-weighting mechanism.

2. Whether augmenting the calibration set with intermediate masked samples at the fragile commit states would be as effective as the proposed position-weighted hidden-state MSE. Could the authors provide additional experimental results to compare this on-policy data augmentation approach against the current weighting scheme?

**Limitations:**

The authors provide an Impact Statement, but the manuscript lacks a dedicated discussion regarding the limitations of the proposed framework.

**Strengths And Weaknesses:**

Strengths:
1. The authors provide a motivating and insightful analysis of dLLM inference behavior, correctly identifying that quantization errors lead to irreversible and cumulative decision flips.

2. The paper is well-rounded, offering not only strong empirical improvements but also theoretical justifications for the proposed method under mild assumptions.

Weaknesses:
1. The paper validates the position-weighted hidden-state MSE solely by optimizing FlatQuant's affine transformation parameters, lacking broader validation. Applying this weighted metric as a drop-in replacement within other diverse PTQ methods (e.g., Qdrop, Percentile) would better substantiate its generalizability.

2. The calibration phase uses only full real tokens instead of intermediate masked samples, creating a domain mismatch between calibration and inference. It is unclear if the proposed position-weighting strategy outperforms a simpler baseline that directly includes actual masked samples from fragile commit states in the calibration set.

3. The paper lacks a sensitivity analysis regarding the relative ratio of the weighting coefficients, specifically $\lambda_0$ versus $\lambda_1$ in Equation 9.

4. Although the authors theoretically demonstrate that the weighted hidden-state MSE serves as an upper bound for output divergence, the paper lacks the theoretical or strong empirical evidence to support that layer-wise optimization on a small calibration set can reliably preserve fragile write-frontier decisions across unseen, out-of-domain distributions.

---

> ### Author Rebuttal · Authors · 2026-03-30
>
> **We sincerely appreciate Reviewer 2Sgo’s valuable feedback and constructive comments. Our point-by-point responses are detailed below.**
>
> **W1&Q1 Generalizability beyond FlatQuant (QDrop-style [1] experiments).**
>
> **R1:** To isolate the effectiveness of our weighted objective from the affine transformation used, we applied our position-weighted MSE as a drop-in replacement for the standard MSE in a **QDrop-style setting**. Since QDrop was originally developed for smaller and more conventional encoder-style models, its direct transfer to our bidirectional diffusion LLM is not straightforward. In particular, it suffers from very severe degradation under W4A4, so we report W8A8 results as a more stable and fair test of whether our weighting objective generalizes beyond FlatQuant. Concretely, we directly optimized only the quantization parameters (scales and zero-points) on W8A8 LLaDA-Instruct for fair comparison. The results are listed below (full table in https://anonymous.4open.science/r/16838-FAIR-Calib-48CB/tabs/QDrop_style_quant.md).
> | Method | Avg. |
> |---|---:|
> | Real-valued | 73.81 |
> | W8A8 QDrop-style | 70.66 |
> | W8A8 QDrop-style + position-weighted MSE | 73.17 |
> | **W8A8 FAIR-Calib** | **73.81** |
>
> The results show a significant improvement of **+2.51** points, nearly closing the gap to FP. This demonstrates that the position-weighting mechanism is a versatile, standalone objective that benefits diverse PTQ optimization targets, not just affine factors.
>
>
> **W2&Q2: Teacher-forced weighting vs. Masked-sample augmentation.**
>
> **R2:** We compared our teacher-forced weighting against two data-augmentation baselines (using W4A4 LLaDA-Instruct) that include intermediate masked states in the calibration set:
>
> - **Random Mask**: Creating pseudo-intermediate states by randomly masking tokens based on sampled diffusion steps.
> - **On-policy (Fragile)**: Running teacher rollouts and selecting only the single most fragile state (lowest commit confidence) per sample.
>
> The results are shown below (full table in https://anonymous.4open.science/r/16838-FAIR-Calib-48CB/tabs/Mask_aug.md).
> | Method | Avg. |
> |---|---:|
> | Real-valued | 74.36 |
> | W4A4 random_mask | 72.60 |
> | W4A4 on-policy | 72.85 |
> | **W4A4 Fair-Calib** | **72.96** |
>
> While augmentation helps, it underperforms FAIR-Calib. We hypothesize this is because weighting provides a "continuous" importance signal across all positions, whereas augmentation only implicitly emphasizes positions by their presence in the data. Furthermore, FAIR-Calib remains highly efficient; an "all-step" on-policy augmentation would scale calibration costs with the number of decoding steps.
>
> **W3: Sensitivity to $\lambda_0$ and $\lambda_1$.**
>
> **R3:** We conducted a ratio sweep for the frontier-hit ($\lambda_0$) and reliability ($\lambda_1$) coefficients on LLaDA-Instruct (W4A4):
> | $\lambda_0$ | $\lambda_1$ | WinoGrande |
> |---|---|---:|
> | 2 | 1 | 75.93 |
> | 1 | 1 | **76.01** |
> | 1 | 2 | 76.00 |
> | 0 | 1 | 75.72 |
> | 1 | 0 | 75.83 |
>
> The results above (also in https://anonymous.4open.science/r/16838-FAIR-Calib-48CB/tabs/lambda_ratio.md) indicate that performance is fairly stable over a moderate range of ratios, with the default 1:1 setting performing near the best, while removing either term leads to a clear drop. We will include the full results in the anonymous supplementary material and revision.
>
> **W4: Preserving fragile write-frontier decisions**
>
> **R4:** We appreciate this concern and agree that this link should be made more explicit. In fact, the current paper already provides several pieces of relevant evidence: Sec. 5.4 further shows reduced teacher-forced commit-step flips and suppressed false-commit error amplification under FAIR-Calib.
>
> To further strengthen this point, we include visualizaiton of two decision-margin diagnostics (see https://anonymous.4open.science/r/16838-FAIR-Calib-48CB/imgs/decision_margin/frontier_margin_at_commit_steps.png and https://anonymous.4open.science/r/16838-FAIR-Calib-48CB/imgs/decision_margin/CDF_of_frontier_margin.png): (i) the CDF of frontier margin at commit steps, and (ii) the average frontier margin at commit steps. Both show better margin preservation under FAIR-Calib; in particular, the average frontier margin increases from 6.56±0.62 for the unweighted baseline to 7.23±0.64 under FAIR-Calib, providing more direct evidence that the proposed weighted objective better preserves fragile write-relevant decisions after quantization.
>
> Summary of revisions:
> - Added QDrop-style results to prove generalizability.
> - Added data-augmentation baselines to justify the weighting strategy.
> - Included a $\lambda$ sensitivity analysis.
> - Added decision-margin diagnostics to the appendix.
>
> We believe these additions significantly strengthen the paper’s claims. We will include the full tables and analysis in the final revision.
>
> [1] Wang, X., et al. QDrop: Randomly Dropping Quantization for Extremely Low-bit Post-Training Quantization. In ICLR, 2022.

---

> > ### Author Rebuttal · Reviewer_2Sgo · 2026-04-02
> >
> > The authors have addressed most of my major concerns in the rebuttal with concrete additional evidence, including QDrop-style generalization experiments, comparisons against masked-sample augmentation baselines, a λ0/λ1sensitivity analysis, and more direct diagnostics on preserving fragile frontier decisions. These additions substantially strengthen the paper’s empirical support and make the central claims more convincing, so I am willing to raise my score.

---

### Decision · Program_Chairs · 2026-04-30

**Decision:**

Accept (regular)

**Comment:**

This paper studies post-training quantization of Diffusion Large Language Models and proposes a method called Frontier-Aware Instability-Reweighted Calibration. There is substantial disagreement among the reviewers. In particular, Reviewer RTjF raised two major concerns:

1. the paper allegedly plagiarizes DLLMQuant and QuantLLM, from the problem definition to the proposed two-stage framework;
2. the theoretical analysis is invalid because the KL decomposition relies on false assumptions. Specifically, the reviewer argued that the commit policy is assumed to be model-independent (i.e., random commit), whereas in reality commit policies are model-dependent. The reviewer also pointed out that the paper assumes the quantized model and the full-precision model share identical commit sets $C_t$, while quantization may significantly distort the distributions and thus lead to quite different $C_t$.

The other three reviewers are generally supportive of the paper (4, 5, 4).

Because of these disputes, I carefully read the submission myself, along with the other two papers mentioned by the reviewer. First, regarding whether the problem definition is plagiarized: the motivation of this submission is that low-bit quantization can flip borderline write decisions, and the resulting errors are then locked in and amplified across refinement steps. In other words, the authors argue that in post-training quantization for DLLMs, quantization error can easily change borderline decisions at the write frontier, after which the errors become permanently locked in and amplified.

By contrast, the statement in DLLMQuant, which the reviewer also quoted, is more general and reflects a common description of DLLMs: DLLMs decode a fixed-length sequence initialized entirely with mask tokens through multiple iterations, and this iterative process leads to divergent input distributions across time steps. Based on this observation, DLLMQuant emphasizes the temporal accumulation of quantization errors. In fact, QuantLLM seems conceptually closer to DLLMQuant, since it also discusses how timestep progression and mask schedules shift the activation distribution, and how quantization errors accumulate across denoising steps and become larger at later stages.

So, while quantization errors accumulating across denoising steps is indeed a common starting point shared by all three papers, this is also a fairly general property of DLLMs and, in my view, does not constitute plagiarism. The submission here is more specific and narrower in scope, focusing explicitly on borderline decisions at the write frontier. Although the practical effect may ultimately still be to better handle these difficult boundary tokens, I do not think it is fair to characterize the motivation itself as plagiarism. The scope and granularity of the problem definition are meaningfully different.

As for whether the method is plagiarized, the main point of dispute concerns the fact that this submission proposes a two-stage PTQ framework for DLLMs: in Stage 1, it probes a full-precision teacher to estimate a frontier-aware, reliability-gated position prior. In Stage 2, it performs off-policy, layer-wise teacher-forcing calibration using a weighted hidden-state MSE. This was compared by the reviewer to DLLMQuant, which uses 1) Temporal-Mask Adaptive Sampling, 2) Interaction-Aware Activation Quantization, and 3) Certainty-Guided Quantization. Although both works broadly aim to identify important positions or tokens for calibration, the actual methods are clearly different, and I do not think the submission can reasonably be considered plagiarized at the method level.

Regarding the theoretical analysis, the main controversy centers on the assumption that, under model-independent random commits and mild smoothness, the overall KL divergence between the full-precision model output distribution and the quantized model output distribution can be upper-bounded by a sum of local errors over time steps and token positions, and that each local error can in turn be controlled by the squared hidden-state difference. This then motivates minimizing a weighted hidden-state MSE as a proxy for minimizing the final output-distribution mismatch.

Here, I do think the assumptions are somewhat questionable in practice, especially the assumption of model-independent random commits, which appears rather strong and somewhat unrealistic. Reviewer j4EW also raised this issue. The authors did not directly address it in the rebuttal, and their later response to the Program Chairs also did not provide a real explanation for this point. By contrast, another modest assumption that under sufficient smoothness, the quality gap between the quantized model and the full-precision model may be reflected through hidden-state differences seems acceptable. In other words, the mild smoothness assumption is at least somewhat defensible. The assumption of model-independent random commits, however, seems much less convincing. Of course, these theoretical imperfections may not necessarily invalidate the empirical effectiveness of the method, as we all know, many useful ideas in deep learning are not yet fully captured by theory.

After carefully considering the reviewer comments, the authors' clarifications, and my own reading of the paper, my conclusion is as follows. I do not think the paper should be considered plagiarized, although it would certainly be beneficial for the authors to cite the two related papers more explicitly and discuss the similarities and differences more clearly. The theoretical analysis contains some worthwhile insights, but its assumptions are indeed somewhat strained and incomplete, and the authors should revise this part more carefully in a future version. There are also several presentation issues that should be polished, for example, there are citation-formatting problems around line 14 and line 319, and parts of Figure 2 appear to be screenshots and are noticeably blurrier than the rest of the paper.

Taking all of this into account, my recommendation is weak accept.